# Learning via Wasserstein-Based High Probability Generalisation Bounds

**Paul Viallard**[*]
Inria, CNRS, Ecole Normale Supérieure,
PSL Research University, Paris, France
`paul.viallard@inria.fr`

**Maxime Haddouche**[*]
Inria, University College London and
Université de Lille, France
`maxime.haddouche@inria.fr`

**Umut Şimşekli**
Inria, CNRS, Ecole Normale Supérieure
PSL Research University, Paris, France
`umut.simsekli@inria.fr`

**Benjamin Guedj**
Inria and University College London,
France and UK
`benjamin.guedj@inria.fr`

## Abstract

Minimising upper bounds on the population risk or the generalisation gap has been widely used in structural risk minimisation (SRM) – this is in particular at the core of PAC-Bayesian learning. Despite its successes and unfailing surge of interest in recent years, a limitation of the PAC-Bayesian framework is that most bounds involve a Kullback-Leibler (KL) divergence term (or its variations), which might exhibit erratic behavior and fail to capture the underlying geometric structure of the learning problem – hence restricting its use in practical applications. As a remedy, recent studies have attempted to replace the KL divergence in the PAC-Bayesian bounds with the Wasserstein distance. Even though these bounds alleviated the aforementioned issues to a certain extent, they either hold in expectation, are for bounded losses, or are nontrivial to minimize in an SRM framework. In this work, we contribute to this line of research and prove novel Wasserstein distance-based PAC-Bayesian generalisation bounds for both batch learning with independent and identically distributed (*i.i.d.*) data, and online learning with potentially non-*i.i.d.* data. Contrary to previous art, our bounds are stronger in the sense that *(i)* they hold with high probability, *(ii)* they apply to unbounded (potentially heavy-tailed) losses, and *(iii)* they lead to optimizable training objectives that can be used in SRM. As a result we derive novel Wasserstein-based PAC-Bayesian learning algorithms and we illustrate their empirical advantage on a variety of experiments.

## 1 Introduction

Understanding generalisation is one of the main challenges in statistical learning theory, and even more so in modern machine learning applications. Typically, a *learning problem* is described by a tuple $(\mathcal{H}, \mathcal{Z}, \ell)$ consisting of a hypothesis (or predictor) space $\mathcal{H}$, a data space $\mathcal{Z}$, and a loss function $\ell : \mathcal{H} \times \mathcal{Z} \to \mathbb{R}$. The goal is to estimate the *population risk* of a given hypothesis $h$, defined as $\mathrm{R}_\mu(h) = \mathbb{E}_{\mathbf{z} \sim \mu}[\ell(h, \mathbf{z})]$, where $\mu$ denotes the unknown *data distribution* over $\mathcal{Z}$.

As $\mu$ is not known, in practice, a hypothesis $h$ is usually built by (approximately) minimising the *empirical risk*, given by $\hat{\mathrm{R}}_{\mathcal{S}}(h) = \frac{1}{m} \sum_{i=1}^{m} \ell(h, \mathbf{z}_i)$, where $\mathcal{S} = \{\mathbf{z}_i \in \mathcal{Z}\}_{i=1}^{m}$ is a dataset of $m$ data points, independent and identically distributed (*i.i.d.*) from $\mu$. We define the generalisation gap of a hypothesis $h$ as $\hat{\mathrm{R}}_{\mathcal{S}}(h) - \mathrm{R}_\mu(h)$.

---

[*]The authors contributed equally to this work

37th Conference on Neural Information Processing Systems (NeurIPS 2023).

Developing upper bounds on the generalisation gap, *i.e.*, *generalisation bounds* has been a long-standing topic in statistical learning. While a plethora of techniques have been introduced, the PAC-Bayesian framework has gained significant traction over the past two decades to provide non-vacuous generalisation guarantees for complex structures such as neural networks during the training phase (see DR17, PORPH⁺21, PRSS21, among others). In these works, the bounds are also used to derive learning algorithms by minimising the right-hand side of a given bound. Beyond neural networks, the flexibility of PAC-Bayes learning makes it a useful toolbox to derive both theoretical results and practical algorithms in various learning fields such as reinforcement learning [FP10], online learning [HG22], multi-armed bandits [SLST⁺11, SLCB⁺12, SAC23], meta-learning [AM18, FM21, RFJK21, RJFK22, DCL⁺21] to name but a few. The PAC-Bayesian bounds focus on a *randomised* setting where the hypothesis is drawn from a *data-dependent* distribution $\rho \in \mathcal{M}(\mathcal{H})$, where $\mathcal{M}(\mathcal{H})$ denotes the set of probability distributions defined on $\mathcal{H}$. A classical PAC-Bayesian result is [Mau04, Theorem 5] (the so-called McAllester bound), which states that, with probability at least $1 - \delta$, for any posterior distribution $\rho \in \mathcal{M}(\mathcal{H})$,

$$\mathbb{E}_{h \sim \rho} \left[ \mathbf{R}_\mu(h) - \hat{\mathbf{R}}_\mathcal{S}(h) \right] \leq \sqrt{\frac{\mathrm{KL}(\rho \| \pi) + \ln \frac{2\sqrt{m}}{\delta}}{2m}},$$

where $\pi \in \mathcal{M}(\mathcal{H})$ is any data-free distribution and KL denotes the Kullback-Leibler divergence. In analogy with Bayesian statistics, $\pi$ is often called the *prior*, and $\rho$ is called the *posterior* - we refer to [Gue19] for a discussion on these terms.

While PAC-Bayesian bounds remain nowadays of the utmost interest to explain generalisation in various learning problems [LGGL19, MGG19, NGG20, MGW20, BG21, HGRS21, ZVM⁺21, BG22b, BG22a, BZG22, CSDG22, LFK⁺22, FRKP22, BG23, SAC23, RACA23], they mostly rely on the KL divergence or variants which causes two main limitations: *(i)* as illustrated in the generative modeling literature, the KL divergence does not incorporate the underlying geometry or topology of the data space $\mathcal{Z}$, hence can behave in an erratic way [ACB17], *(ii)* the KL divergence and its variants require the posterior $\rho$ to be absolutely continuous with respect to the prior $\pi$. However, recent studies [CDE⁺21] have shown that, in stochastic optimisation, the distribution of the iterates, which is the natural choice for the posterior, can converge to a *singular distribution*, which does not admit a density with respect to the Lebesgue measure. Moreover, the structure of the singularity (*i.e.*, the *fractal dimension* of $\rho$) depends on the data sample $\mathcal{S}$ [CDE⁺21]. Hence, in such a case, it would not be possible to find a suitable prior $\pi$ that can dominate $\rho$ for almost every $\mathcal{S} \sim \mu^m$, which will trivially make $\mathrm{KL}(\rho \| \pi) = +\infty$ and the generalisation bound vacuous.

Some works have focused on replacing the Kullback-Leibler divergence with more general divergences in PAC-Bayes [AG18, OH21, PWG22], although the problems arising from the presence of the KL divergence in the generalisation bounds are actually not specific to PAC-Bayes: information-theoretic bounds [GMGA17, XR17, RZ20] also suffer from similar issues as they are based on a mutual information term, which is the KL divergence between two distributions. In this context, as a remedy to these issues introduced by the KL divergence, [ZLT18, WDFC19, GBTS21, LN22] proved analogous bounds that are based on the *Wasserstein distance*, which arises from the theory of optimal transport [Mon81]. As the Wasserstein distance inherits the underlying geometry of the data space and does not require absolute continuity, it circumvents the problems introduced by the KL divergence. Yet, these bounds hold only in expectation, *i.e.*, none of these bounds is holding with high probability over the random choice of the learning sample $\mathcal{S} \sim \mu^m$.

In the context of PAC-Bayesian learning, the recent works [AEMM22, CL21] incorporated Wasserstein distances as a complexity measure and proved generalisation bounds based on the Wasserstein distance. More precisely, [AEMM22] proved a high-probability generic PAC-Bayesian bound for bounded losses depending on an integral probability metric [Mü97], which contains the Wasserstein distance as a special case. On the other hand, [CL21] exploited PAC-Bayesian tools to obtain learning strategies with their associated regret bounds based on the Wasserstein distance for the *online learning* setting while requiring a finite hypothesis space and do not deal with generalisation.

**Contributions.** The theoretical understanding of the high-probability generalisation bounds based on the Wasserstein distance is still limited. The aim of this paper is not only to prove generalisation bounds (for different learning settings) based on the optimal transport theory but also to propose new learning algorithms derived from our theoretical results.

*(i)* Using the supermartingale toolbox introduced in [HG23a, CWR23], we prove in Section 3.1, novel PAC-Bayesian bounds based on the Wasserstein distance for *i.i.d.* data. While [AEMM22] proposed a McAllester-like bound for bounded losses, we propose a Catoni-like bound (see *e.g.*, ARC16, Theorem 4.1) valid for heavy-tailed losses with bounded order 2 moments. This assumption is less restrictive than assuming subgaussian or bounded losses, which are at the core of many PAC-Bayes results. This assumption also covers distributions beyond subgaussian or subexponential ones (*e.g.*, gamma distributions with a scale smaller than 1, which have an infinite exponential moment).

*(ii)* We provide in Section 3.2 the first generalisation bounds based on Wasserstein distances for the online PAC-Bayes framework of [HG22]. Our results are, again, Catoni-like bounds and hold for heavy-tailed losses with bounded order 2 moments. Previous work [CL21] already provided online strategies mixing PAC-Bayes and Wasserstein distances. However, their contributions focus on the best deterministic strategy, regularised by a Wasserstein distance, with respect to the deterministic notion of regret. Our results differ significantly as we provide the best-regularised strategy (still in the sense of a Wasserstein term) with respect to the notion of generalisation, which is new.

*(iii)* As our bounds are linear with respect to Wasserstein terms (contrary to those of AEMM22), they are well suited for optimisation procedures. Thus, we propose the first PAC-Bayesian learning algorithms based on Wasserstein distances instead of KL divergences. For the first time, we design PAC-Bayes algorithms able to output deterministic predictors (instead of distributions over all $\mathcal{H}$) designed from deterministic priors. This is due to the ability of the Wasserstein distance to measure the discrepancy between Dirac distributions. We then instantiate those algorithms in Section 4 on various datasets, paving the way to promising practical developments of PAC-Bayes learning.

To sum up, we highlight two benefits of PAC-Bayes learning with Wasserstein distance. First, it ships with sound theoretical results exploiting the geometry of the predictor space, holding for heavy-tailed losses. Such a weak assumption on the loss extends the usefulness of PAC-Bayes with Wasserstein distances to a wide range of learning problems, encompassing bounded losses. Second, it allows us to consider deterministic algorithms (*i.e.*, sampling from Dirac measures) designed with respect to the notion of generalisation: we showcase their performance in our experiments.

**Outline.** Section 2 describes our framework and background, Section 3 contains our new theoretical results and Section 4 gathers our experiments. Appendix A gathers supplementary discussion, Appendix B contains all proofs of our claims, and Appendix C provides insights into our practical results as well as additional experiments.

## 2 Our framework

**Framework.** We consider a Polish predictor space $\mathcal{H}$ equipped with a distance $d$ and a $\sigma$-algebra $\Sigma_{\mathcal{H}}$, a data space $\mathcal{Z}$, and a loss function $\ell : \mathcal{H} \times \mathcal{Z} \to \mathbb{R}$. In this work, we consider Lipschitz functions with respect to $d$. We also associate a filtration $(\mathcal{F}_i)_{i \geq 1}$ adapted to our data $(\mathbf{z}_i)_{i=1,\dots,m}$, and we assume that the dataset $\mathcal{S}$ follows the distribution $\mathcal{D}$. In PAC-Bayes learning, we construct a data-driven posterior distribution $\rho \in \mathcal{M}(\mathcal{H})$ with respect to a prior distribution $\pi$.

**Definitions.** For all $i$, we denote by $\mathbb{E}_i[\cdot]$ the conditional expectation $\mathbb{E}[\cdot \mid \mathcal{F}_i]$. In this work, we consider data-dependent priors. A stochastic kernel is a mapping $\pi : \cup_{m=1}^{\infty} \mathcal{Z}^m \times \Sigma_{\mathcal{H}} \to [0, 1]$ where *(i)* for any $B \in \Sigma_{\mathcal{H}}$, the function $\mathcal{S} \mapsto \pi(\mathcal{S}, B)$ is measurable, *(ii)* for any dataset $\mathcal{S}$, the function $B \mapsto \pi(\mathcal{S}, B)$ is a probability measure over $\mathcal{H}$.

In what follows, we consider two different learning paradigms: *batch learning*, where the dataset is directly available, and *online learning*, where data streams arrive sequentially.

**Batch setting.** We assume the dataset $\mathcal{S}$ to be *i.i.d.*, so there exists a distribution $\mu$ over $\mathcal{Z}$ such that $\mathcal{D} = \mu^m$. We then define, for a given $h \in \mathcal{H}$, the *risk* to be $R_\mu := \mathbb{E}_{\mathbf{z} \sim \mu}[\ell(h, \mathbf{z})]$ and its empirical counterpart $\hat{R}_{\mathcal{S}} := \frac{1}{m} \sum_{i=1}^{m} \ell(h, \mathbf{z}_i)$. Our results aim to bound the *expected generalisation gap* defined by $\mathbb{E}_{h \sim \rho}[R_\mu(h) - \hat{R}_{\mathcal{S}}(h)]$. We assume that the dataset $\mathcal{S}$ is split into $K$ disjoint sets $\mathcal{S}_1, \dots, \mathcal{S}_K$. We consider $K$ stochastic kernels $\pi_1, \dots, \pi_K$ such that for any $\mathcal{S}$, the distribution $\pi_i(\mathcal{S}, .)$ *does not* depend on $\mathcal{S}_i$.

**Online setting.** We adapt the online PAC-Bayes framework of [HG22]. We assume that we have access to a stream of data $\mathcal{S} = (\mathbf{z}_i)_{i=1,\dots,m}$, arriving sequentially, with no assumption on $\mathcal{D}$. In online PAC-Bayes, the goal is to define a posterior sequence $(\rho_i)_{i\geq 1}$ from a prior sequence $(\pi_i)_{i\geq 1}$, which can be data-dependent. We define an *online predictive sequence* $(\pi_i)_{i=1\cdots m}$ satisfying: *(i)* for all $i$ and dataset $\mathcal{S}$, the distribution $\pi_i(S,.)$ is $\mathcal{F}_{i-1}$ measurable and *(ii)* there exists $\pi_0$ such that for all $i \geq 1$, we have $\pi_i(S,.) \gg \pi_0$. This last condition covers, in particular, the case where $\mathcal{H}$ is an Euclidean space and for any $i$, the distribution $\pi_{i,\mathcal{S}}$ is a Dirac mass. All of those measures are uniformly continuous with respect to any Gaussian distribution.

**Wasserstein distance.** In this paper, we focus on the Wasserstein distance of order 1 (*a.k.a.*, Earth Mover's distance) introduced by [Kan60] in the optimal transport literature. Given a distance $d : \mathcal{A} \times \mathcal{A} \to \mathbb{R}$ and a Polish space $(\mathcal{A}, d)$, for any probability measures $\alpha$ and $\beta$ on $\mathcal{A}$, the Wasserstein distance is defined by

$$\mathrm{W}(\alpha, \beta) := \inf_{\gamma \in \Gamma(\alpha, \beta)} \left\{ \mathop{\mathbb{E}}_{(a,b)\sim\gamma} d(a, b) \right\}, \tag{1}$$

where $\Gamma(\alpha, \beta)$ is the set of joint probability measures $\gamma \in \mathcal{M}(\mathcal{A}^2)$ such that the marginals are $\alpha$ and $\beta$. The Wasserstein distance aims to find the probability measure $\gamma \in \mathcal{M}(\mathcal{A}^2)$ minimising the expected cost $\mathbb{E}_{(a,b)\sim\gamma} d(a, b)$. We refer the reader to [Vil09, PC19] for an introduction to optimal transport.

## 3 Wasserstein-based PAC-Bayesian generalisation bounds

We present novel high-probability PAC-Bayesian bounds involving Wasserstein distances instead of the classical Kullback-Leibler divergence. Our bounds hold for heavy-tailed losses (instead of classical subgaussian and subexponential assumptions), extending the remits of [AEMM22, Theorem 11]. We exploit the supermartingale toolbox, recently introduced in PAC-Bayes framework by [HG23a, CWR23, JJKO23], to derive bounds for both batch learning (Theorems 1 and 2) and online learning (Theorems 3 and 4).

### 3.1 PAC-Bayes for batch learning with *i.i.d.* data

In this section, we use the batch setting described in Section 2. We state our first result, holding for heavy-tailed losses admitting order 2 moments. Such an assumption is in line, for instance, with reinforcement learning with heavy-tailed reward (see, *e.g.*, LZ11, LWHZ19, ZS21).

**Theorem 1.** *We assume the loss $\ell$ to be L-Lipschitz. Then, for any $\delta \in (0, 1]$, for any sequence of positive scalar $(\lambda_i)_{i\in\{1,\dots,K\}}$, with probability at least $1 - \delta$ over the sample $\mathcal{S}$, the following holds for the distributions $\pi_{i,\mathcal{S}} := \pi_i(\mathcal{S},.)$ and for any $\rho \in \mathcal{M}(\mathcal{H})$:*

$$\mathop{\mathbb{E}}_{h\sim\rho} \left[ R_\mu(h) - \hat{R}_\mathcal{S}(h) \right]$$
$$\leq \sum_{i=1}^{K} \frac{2|\mathcal{S}_i|L}{m} \mathrm{W}(\rho, \pi_{i,\mathcal{S}}) + \frac{1}{m} \sum_{i=1}^{K} \frac{\ln\left(\frac{K}{\delta}\right)}{\lambda_i} + \frac{\lambda_i}{2} \left( \mathop{\mathbb{E}}_{h\sim\pi_{i,\mathcal{S}}} \left[ \hat{V}_{|\mathcal{S}_i|}(h) + V_{|\mathcal{S}_i|}(h) \right] \right),$$

*where $\pi_{i,\mathcal{S}}$ does not depend on $\mathcal{S}_i$. Also, for any $i, |S_i|$, we have $\hat{V}_{|\mathcal{S}_i|}(h) = \sum_{\mathbf{z}\in\mathcal{S}_i} (\ell(h, \mathbf{z}) - R_\mu(h))^2$ and $V_{|\mathcal{S}_i|}(h) = \mathbb{E}_{\mathcal{S}_i} \left[ \hat{V}_{|\mathcal{S}_i|}(h) \right]$.*

The proof is deferred to Appendix B.1. While Theorem 1 holds for losses taking values in $\mathbb{R}$, many learning problems rely in practice on more constrained losses. This loss can be bounded as in the case of, *e.g.*, supervised learning or the multi-armed bandit problem [Sli19], or simply non-negative as in regression problems involving the quadratic loss (studied, for instance, in Cat16, CG17). Using again the supermartingale toolbox, we prove in Theorem 2 a tighter bound holding for heavy-tailed non-negative losses.

**Theorem 2.** *We assume our loss $\ell$ to be non-negative and L-Lipschitz. We also assume that, for any $1 \leq i \leq K$, for any dataset$\mathcal{S}$, we have $\mathbb{E}_{h\sim\pi_i(.,\mathcal{S}),z\sim\mu} \left[ \ell(h, z)^2 \right] \leq 1$ (bounded order 2 moments for*

priors*). Then, for any $\delta \in (0, 1]$, with probability at least $1 - \delta$ over the sample $\mathcal{S}$, the following holds for the distributions $\pi_{i,\mathcal{S}} := \pi_i(\mathcal{S}, .)$ and for any $\rho \in \mathcal{M}(\mathcal{H})$:*

$$\underset{h \sim \rho}{\mathbb{E}} \left[ R_\mu(h) - \hat{R}_\mathcal{S}(h) \right] \leq \sum_{i=1}^{K} \frac{2|\mathcal{S}_i|L}{m} W(\rho, \pi_{i,\mathcal{S}}) + \sum_{i=1}^{K} \sqrt{\frac{2|\mathcal{S}_i| \ln \frac{K}{\delta}}{m^2}},$$

*where $\pi_{i,\mathcal{S}}$ does not depend on $\mathcal{S}_i$.*

Note that when the loss function takes values in $[0, 1]$, an alternative strategy allows tightening the last term of the bound by a factor $^1/_2$. This result is rigorously stated in Theorem 6 of Appendix B.3.

**High-level ideas of the proofs.** Theorems 1 and 2 are structured around two tools. First, we exploit the Kantorovich-Rubinstein duality [Vil09, Remark 6.5] to replace the change of measure inequality [Csi75, DV76]; this allows us to consider a Wasserstein distance instead of a KL term. Then, we exploit the supermartingales used in [HG23a, CWR23] alongside Ville's inequality (instead of Markov's one) to obtain a high probability bound holding for heavy-tailed losses. Combining those techniques provides our PAC-Bayesian bounds.

**Analysis of our bounds.** Our results hold for Lipschitz losses and allow us to consider heavy-tailed losses with bounded order 2 moments. While such an assumption on the loss is more restrictive than in classical PAC-Bayes, allowing heavy-tailed losses is strictly less restrictive. While Theorem 1 is our most general statement, Theorem 2 allows recovering a tighter result (without empirical variance terms) for non-negative heavy-tailed losses. An important point is that the variance terms are considered with respect to the prior distributions $\pi_{i,\mathcal{S}}$ and not $\rho$ as in [HG23a, CWR23]. This is crucial as these papers rely on the implicit assumption of order 2 moments, holding uniformly for all $\rho \in \mathcal{M}(\mathcal{H})$, while we only require this assumption for the prior distributions $(\pi_{i,\mathcal{S}})_{i=1,...,K}$. Such an assumption is in line with the PAC-Bayesian literature, which often relies on bounding an averaged quantity with respect to the prior. This strength is a consequence of the Kantorovich-Rubinstein duality. To illustrate this, consider *i.i.d.* data with distribution $\mu$ admitting a finite variance bounded by $V$ and the loss $\ell(h, z) = |h - z|$ where both $h$ and $z$ lie in the real axis. Notice that in this particular case, we can imagine that $z$ is a data point and $h$ is a hypothesis outputting the same scalar for all data. To satisfy the assumption of Theorem 2, it is enough, by Cauchy Schwarz, to satisfy $\mathbb{E}_{h \sim \pi_{i,\mathcal{S}}, z \sim \mathcal{S}}[\ell(h, z)^2] \leq \mathbb{E}[h^2] + 2V \mathbb{E}[|h|] + V^2 \leq 1$ for all $\pi_{i,\mathcal{S}}$. On the contrary, [HG23a, CWR23] would require this condition to hold for all $\rho$, which is more restrictive. Finally, an important point is that our bound allows us to consider Dirac distributions with disjoint support as priors and posteriors. On the contrary, KL divergence forces us to consider a non-Dirac prior for our bound to be non-vacuous. This allows us to retrieve a uniform-convergence bound described in Corollary 7.

**Role of data-dependent priors.** Theorems 1 and 2 allow the use of prior distributions depending possibly on a fraction of data. Such a dependency is crucial to control our sum of Wasserstein terms as we do not have an explicit convergence rate. For instance, for a fixed $K$, consider a compact predictor space $\mathcal{H}$, a bounded loss and the *Gibbs posterior* defined as $d\rho(h) \propto \exp\left(-\lambda \hat{R}_\mathcal{S}(h)\right) dh$ where $\lambda > 0$. Also define for any $i$ and $\mathcal{S}$, the distribution $d\pi_{i,\mathcal{S}}(h) \propto \exp\left(-\lambda R_{\mathcal{S}/\mathcal{S}_i}(h)\right) dh$. Then, by the law of large numbers, when $m$ goes to infinity, for any $h$, both $R_S(h)$ and $(R_{\mathcal{S}/\mathcal{S}_i}(h))_{i=1,...,m}$ converge to $R_\mu(h)$. This ensures, alongside with the dominated convergence theorem, that for any $i$, the Wasserstein distance $W(\rho, \pi_{i,\mathcal{S}})$ goes to zero as $m$ goes to infinity.

**Comparison with the literature.** [AEMM22, Theorem 11] establishes a PAC-Bayes bound with Wasserstein distance valid for bounded losses being Lipschitz with high probability. While we circumvent the first assumption, the second one is less restrictive than actual Lipschitzness and can also be used in our setting. Also [AEMM22, Theorem 12] proposes an explicit convergence for finite predictor classes. We show in Appendix A that we are also able to recover such a convergence.

**Towards new PAC-Bayesian algorithms.** From Theorem 2, we derive a new PAC-Bayesian algorithm for Lipschitz non-negative losses:

$$\underset{\rho \in \mathcal{M}(\mathcal{H})}{\arg\min} \underset{h \sim \rho}{\mathbb{E}} \left[ \hat{R}_\mathcal{S}(h) \right] + \sum_{i=1}^{K} \frac{2|\mathcal{S}_i|L}{m} W(\rho, \pi_{i,\mathcal{S}}). \tag{2}$$

Equation (2) uses Wasserstein distances as regularisers and allows the use of multiple priors. We compare ourselves to the classical PAC-Bayes algorithm derived from [Cat07, Theorem 1.2.6] (which

leads to Gibbs posteriors):

$$\operatorname*{argmin}_{\rho \in \mathcal{M}(\mathcal{H})} \mathop{\mathbb{E}}_{h \sim \rho} \left[ \hat{R}_{\mathcal{S}}(h) \right] + \frac{\mathrm{KL}(\rho, \pi)}{\lambda}. \tag{3}$$

Considering a Wasserstein distance in Equation (2) makes our algorithm more flexible than in Equation (3), the KL divergence implies absolute continuity *w.r.t.* the prior $\pi$. Such an assumption is not required to use Equation (2) and covers the case of prior Dirac distributions. Finally, Equation (2) relies on a fixed value $K$ whose value is discussed below.

**Role of $K$.** We study the cases $K = 1$, $\sqrt{m}$, and $m$ in Theorem 2. We refer to Appendix A for a detailed treatment. First of all, when $K = 1$, we recover a classical batch learning setting where all data are collected at once. In this case, we have a single Wasserstein with no convergence rate coupled with a statistical ersatz of $\sqrt{\ln(1/\delta)/m}$. However, similarly to [AEMM22, Theorem 12], in the case of a finite predictor class, we are able to recover an explicit convergence rate. The case $K = \sqrt{m}$ provides a tradeoff between the number of points required to have good data-dependent priors (which may lead to a small $\sum_{i=1}^{\sqrt{m}} \mathrm{W}(\rho, \pi_i)$) and the number of sets required to have an explicit convergence rate. Finally, the case $K = m$ leads to a vacuous bound as we have the incompressible term $\sqrt{\ln(m/\delta)}$, which makes the bound vacuous for large values of $m$. This means that the batch setting is not fitted to deal with a data stream arriving sequentially. To mitigate that weakness, we propose in Section 3.2 the first online PAC-Bayes bounds with Wasserstein distances.

### 3.2 Wasserstein-based generalisation bounds for online learning

Here, we use the online setting described in Section 2 and derive the first online PAC-Bayes bounds involving Wasserstein distances in Theorems 3 and 4. Online PAC-Bayes bounds are meant to derive online counterparts of classical PAC-Bayesian algorithms [HG22], where the KL-divergence acts as a regulariser. We show in Theorems 3 and 4 that it is possible to consider online PAC-Bayesian algorithms where the regulariser is a Wasserstein distance, which allows us to optimise on measure spaces without a restriction of absolute continuity.

**Theorem 3.** *We assume our loss $\ell$ to be L-Lipschitz. Then, for any $\delta \in (0, 1]$, with probability at least $1 - \delta$ over the sample $\mathcal{S}$, the following holds for the distributions $\pi_{i,\mathcal{S}} := \pi_i(\mathcal{S}, .)$ and for any sequence $(\rho_i)_{i=1\cdots m} \in \mathcal{M}(\mathcal{H})^m$:*

$$\sum_{i=1}^{m} \mathop{\mathbb{E}}_{h_i \sim \rho_i} \left[ \mathbb{E}[\ell(h_i, \mathbf{z}_i) \mid \mathcal{F}_{i-1}] - \ell(h_i, \mathbf{z}_i) \right] \leq 2L \sum_{i=1}^{m} \mathrm{W}(\rho_i, \pi_{i,\mathcal{S}})$$

$$+ \frac{\lambda}{2} \sum_{i=1}^{m} \mathop{\mathbb{E}}_{h_i \sim \pi_{i,\mathcal{S}}} \left[ \hat{V}_i(h_i, \mathbf{z}_i) + V_i(h_i) \right] + \frac{\ln(1/\delta)}{\lambda},$$

*where for all $i$, $\hat{V}_i(h_i, \mathbf{z}_i) = (\ell(h_i, \mathbf{z}_i) - \mathbb{E}_{i-1}[\ell(h_i, \mathbf{z}_i)])^2$ is the conditional empirical variance at time $i$ and $V_i(h_i) = \mathbb{E}_{i-1}[\hat{V}(h_i, \mathbf{z}_i)]$ is the true conditional variance.*

The proof is deferred to Appendix B.4. We also provide the following bound, being an online analogous of Theorem 2, valid for non-negative heavy-tailed losses.

**Theorem 4.** *We assume our loss $\ell$ to be non-negative and L-Lipschitz. We also assume that, for any $i, \mathcal{S}$, $\mathbb{E}_{h \sim \pi_i(.,\mathcal{S})} \left[ \mathbb{E}_{i-1}[\ell(h, \mathbf{z}_i)^2] \right] \leq 1$ (bounded conditional order 2 moments for priors). Then, for any $\delta \in (0, 1]$, with probability at least $1 - \delta$ over the sample $\mathcal{S}$, any online predictive sequence (used as priors) $(\pi_i)_{i \geq 1}$, we have with probability at least $1 - \delta$ over the sample $S \sim \mu$, the following, holding for the data-dependent measures $\pi_{i,\mathcal{S}} := \pi_i(S, .)$ and any posterior sequence $(\rho_i)_{i \geq 1}$:*

$$\frac{1}{m} \sum_{i=1}^{m} \mathop{\mathbb{E}}_{h_i \sim \rho_i} \left[ \mathbb{E}[\ell(h_i, \mathbf{z}_i) \mid \mathcal{F}_{i-1}] - \ell(h_i, \mathbf{z}_i) \right] \leq \frac{2L}{m} \sum_{i=1}^{m} \mathrm{W}(\rho_i, \pi_{i,\mathcal{S}}) + \sqrt{\frac{2\ln\left(\frac{1}{\delta}\right)}{m}}.$$

The proof is deferred to Appendix B.5.

**Analysis of our bounds.** Theorems 3 and 4 are, to our knowledge, the first results involving Wasserstein distances for online PAC-Bayes learning. They are the online counterpart of Theorems 1 and 2, and the discussion of Section 3.1 about the involved assumptions also apply here. The sum of

Wasserstein distances involved here is a consequence of the online setting and must grow sublinearly for the bound to be tight. For instance, when $(\rho_i = \delta_{h_i})_{i \geq 1}$ is the output of an online algorithm outputting Dirac measures and $\pi_{i,\mathcal{S}} = \rho_{i-1}$, the sum of Wasserstein is exactly $\sum_{i=1}^{m} d(h_i, h_{i-1})$. This sum has to be sublinear for the bound to be non-vacuous, and the tightness depends on the considered learning problem. An analogous of this sum can be found in dynamic online learning [Zin03] where similar sums appear as *path lengths* to evaluate the complexity of the problem.

**Comparison with literature.** We compare our results to existing PAC-Bayes bounds for martingales of [SLCB$^+$12]. [SLCB$^+$12, Theorem 4] is a PAC-Bayes bound for martingales, which controls an average of martingales, similar to our Theorem 1. Under a boundedness assumption, they recover a McAllester-typed bound, while Theorem 1 is more of a Catoni-typed result. Also, [SLCB$^+$12, Theorem 7] is a Catoni-typed bound involving a conditional variance, similar to our Theorem 4. They require to bound uniformly the variance on all the predictor sets, while we only assume averaged variance with respect to priors, which is what we required to perform Theorem 4.

**A new online algorithm.** [HG22] derived from their main theorem, an online counterpart of Equation (3), proving it comes with guarantees. Similarly, we exploit Theorem 4 to derive the online counterpart of Equation (2), from the data-free initialisation $\rho_1$

$$\forall i \geq 1, \ \rho_i \in \underset{\rho \in \mathcal{M}(\mathcal{H})}{\operatorname{argmin}} \ \underset{h \sim \rho}{\mathbb{E}} \left[ \ell(h_i, \mathbf{z}_i) \right] + 2L\mathrm{W}(\rho, \pi_{i,\mathcal{S}}). \tag{4}$$

We highlight the merits of the algorithm defined by Equation (4), alongside with the one from Equation (2), in Section 4.

## 4   Learning via Wasserstein regularisation

Theorems 2 and 4 are designed to be informative on the generalisation ability of a single hypothesis even when Dirac distributions are considered. In particular, our results involve Wasserstein distances acting as regularisers on $\mathcal{H}$. In this section, we show that a Wasserstein regularisation of the learning objective, which comes from our theoretical bounds, helps to better generalise in practice. Inspired by Equations (2) and (4), we derive new PAC-Bayesian algorithms for both batch and online learning involving a Wasserstein distance (see Section 4.1), we describe our experimental framework in Section 4.2 and we present some of the results in Section 4.3. Additional details, experiments, and discussions are gathered in Appendix C due to space constraints. All the experiments are reproducible with the source code provided on GitHub at `https://github.com/paulviallard/NeurIPS23-PB-Wasserstein`.

### 4.1   Learning algorithms

**Classification.** In the classification setting, we assume that the data space $\mathcal{Z} = \mathcal{X} \times \mathcal{Y}$ is composed of a $d$-dimensional *input space* $\mathcal{X} = \{\mathbf{x} \in \mathbb{R}^d \mid \|\mathbf{x}\|_2 \leq 1\}$ and a finite *label space* $\mathcal{Y} = \{1, \ldots, |\mathcal{Y}|\}$ with $|\mathcal{Y}|$ labels. We aim to learn models $h_{\mathbf{w}} : \mathbb{R}^d \to \mathbb{R}^{|\mathcal{Y}|}$ parameterised by a weight vector $\mathbf{w}$ that outputs, given an input $\mathbf{x} \in \mathcal{X}$, a score $h_{\mathbf{w}}(\mathbf{x})[y'] \in \mathbb{R}$ for each label $y'$. This score allows us to assign a label to $\mathbf{x} \in \mathcal{X}$; to check if $h_{\mathbf{w}}$ classifies correctly the example $(\mathbf{x}, y)$, we use the *classification loss* defined by $\ell^c(h_{\mathbf{w}}, (\mathbf{x}, y)) := \mathbb{1}\left[ h_{\mathbf{w}}(\mathbf{x})[y] - \max_{y' \neq y} h_{\mathbf{w}}(\mathbf{x})[y'] \leq 0 \right]$, where $\mathbb{1}$ denotes the indicator function.

**Batch algorithm.** In the batch setting, we aim to learn a parametrised hypothesis $h_{\mathbf{w}} \in \mathcal{H}$ that minimises the population classification risk $\mathfrak{R}_\mu(h_{\mathbf{w}}) = \mathbb{E}_{(\mathbf{x}, y) \sim \mu} \ell^c(h_{\mathbf{w}}, (\mathbf{x}, y))$ that we can only estimate through the empirical classification risk $\mathfrak{R}_{\mathcal{S}}(h_{\mathbf{w}}) = \frac{1}{m} \sum_{i=1}^{m} \ell^c(h_{\mathbf{w}}, (\mathbf{x}_i, y_i))$. To learn the hypothesis, we start from Equation (2), when the distributions $\rho$ and $\pi_1, \ldots, \pi_K$ are Dirac masses, localised at $h_{\mathbf{w}}, h_{\mathbf{w}_1}, \ldots h_{\mathbf{w}_K} \in \mathcal{H}$ respectively. Indeed, in this case, $\mathrm{W}(\rho, \pi_{i,\mathcal{S}}) = d(h_{\mathbf{w}}, h_{\mathbf{w}_i})$ for any $i$. However, the loss $\ell^c(., \mathbf{z})$ is not Lipschitz and the derivatives are zero for all examples $\mathbf{z} \in \mathcal{X} \times \mathcal{Y}$, which prevents its use in practice to obtain such a hypothesis $h_{\mathbf{w}}$. Instead, for the population risk $\mathrm{R}_\mu(h)$ and the empirical risk $\hat{\mathrm{R}}_{\mathcal{S}}(h)$ (in Theorem 2 and Equation (2)), we consider the loss $\ell(h, (\mathbf{x}, y)) = \frac{1}{|\mathcal{Y}|} \sum_{y' \neq y} \max(0, 1 - \eta(h[y] - h[y']))$, which is $\eta$-Lipschitz *w.r.t.* the outputs $h[1], \ldots, h[|\mathcal{Y}|]$. This loss has subgradients everywhere, which is convenient in practice. We go a

step further by *(a)* setting $L = \frac{1}{2}$ and *(b)* adding a parameter $\varepsilon > 0$ to obtain the objective

$$\operatorname*{argmin}_{h_{\mathbf{w}} \in \mathcal{H}} \left\{ \hat{R}_{\mathcal{S}}(h_{\mathbf{w}}) + \varepsilon \left[ \sum_{i=1}^{K} \frac{|\mathcal{S}_i|}{m} d\left(h_{\mathbf{w}}, h_{\mathbf{w}_i}\right) \right] \right\}. \tag{5}$$

To (approximately) solve Equation (5), we propose a two-step algorithm. First, PRIORS LEARN-ING learns $K$ hypotheses $h_{\mathbf{w}_1}, \ldots, h_{\mathbf{w}_K} \in \mathcal{H}$ by minimising the empirical risk via stochastic gradient descent. Second, POSTERIOR LEARNING learns the hypothesis $h_{\mathbf{w}} \in \mathcal{H}$ by minimising the objective associated with Equation (5). More precisely, PRIORS LEARNING outputs the hypotheses $h_{\mathbf{w}_1}, \cdots, h_{\mathbf{w}_K}$, obtained by minimising the empirical risk through mini-batches. Those batches are designed such that for any $i$, the hypothesis $h_{\mathbf{w}_i}$ does not depend on $\mathcal{S}_i$. Then, given $h_{\mathbf{w}_1}, \ldots, h_{\mathbf{w}_K} \in \mathcal{H}$, POSTERIOR LEARNING minimises the objective in Equation (5) with mini-batches. Those algorithms are presented in Algorithm 1 of Appendix C. While $\varepsilon$ is not suggested by Equation (2), it helps to control the impact of the regularisation in practice. Equation (5) then optimises a tradeoff between the empirical risk and the regularisation term $\varepsilon \sum_{i=1}^{K} \frac{|\mathcal{S}_i|}{m} d(h_{\mathbf{w}}, h_{\mathbf{w}_i})$.

**Online algorithm.** Online algorithms output, at each time step $i \in \{1, \ldots, m\}$, a new hypothesis $h_{\mathbf{w}_i}$. From Equation (4), particularised to a sequence of Dirac distributions (localised in $h_{\mathbf{w}_1}, \cdots, h_{\mathbf{w}_K}$), we design a novel online PAC-Bayesian algorithm with a Wasserstein regulariser:

$$\forall i \geq 1, \quad h_i \in \operatorname*{argmin}_{h_{\mathbf{w}} \in \mathcal{H}} \ell(h_{\mathbf{w}}, \mathbf{z}_i) + d\left(h_{\mathbf{w}}, h_{\mathbf{w}_{i-1}}\right) \quad \text{s.t.} \quad d\left(h_{\mathbf{w}}, h_{\mathbf{w}_{i-1}}\right) \leq 1. \tag{6}$$

According to Theorem 4, such an algorithm aims to bound the *population cumulative classification loss* $\mathfrak{C}_\mu = \sum_{i=1}^{m} \mathbb{E}[\ell^c(h_{\mathbf{w}_i}, \mathbf{z}_i) \mid \mathcal{F}_{i-1}]$. Note that we added the constraint $d\left(h_{\mathbf{w}}, h_{\mathbf{w}_{i-1}}\right) \leq 1$ compared to Equation (4). This constraint ensures that the new hypothesis $h_{\mathbf{w}_i}$ is not too far from $h_{\mathbf{w}_{i-1}}$ (in the sense of the distance $\|\cdot\|_2$). Note that the constrained optimisation problem in Equation (6) can be rewritten in an unconstrained form (see [BV04]) thanks to a barrier $B(\cdot)$ defined by $B(a) = 0$ if $a \leq 0$ and $B(a) = +\infty$ otherwise; we have

$$\forall i \geq 1, \quad h_i \in \operatorname*{argmin}_{h_{\mathbf{w}} \in \mathcal{H}} \ell(h_{\mathbf{w}}, \mathbf{z}_i) + d\left(h_{\mathbf{w}}, h_{\mathbf{w}_{i-1}}\right) + B(d\left(h_{\mathbf{w}}, h_{\mathbf{w}_{i-1}}\right) - 1). \tag{7}$$

When solving the problem in Equation (7) is not feasible, we approximate it with a log barrier of [KDY$^+$22] (suitable in a stochastic gradient setting); given a parameter $t > 0$, the log barrier extension is defined by $\hat{B}(a) = -\frac{1}{t}\ln(-a)$ if $a \leq -\frac{1}{t^2}$ and $\hat{B}(a) = ta - \frac{1}{t}\ln(\frac{1}{t^2}) + \frac{1}{t}$ otherwise. We present in Appendix C Algorithm 2 that aims to (approximately) solve Equation (7). To do so, for each new example $(\mathbf{x}_i, y_i)$, the algorithm runs several gradient descent steps to optimise Equation (7).

### 4.2 Experimental framework

In this part, we assimilate the predictor space $\mathcal{H}$ to the parameter space $\mathbb{R}^d$. Thus, the distance $d$ is the Euclidean distance between two parameters: $d\left(h_{\mathbf{w}}, h_{\mathbf{w}'}\right) = \|\mathbf{w} - \mathbf{w}'\|_2$. This implies that the Lipschitzness of $\ell$ has to be taken *w.r.t.* $\mathbf{w}$ instead of $h_{\mathbf{w}}$.

**Models.** We consider that the models are either linear or neural networks (NN). Linear models are defined by $h_{\mathbf{w}}(\mathbf{x}) = W\mathbf{x} + b$, where $W \in \mathbb{R}^{|\mathcal{Y}| \times d}$ is the weight matrix, $b \in \mathbb{R}^{|\mathcal{Y}|}$ is the bias, and $\mathbf{w} = \text{vec}(\{W, b\})$ its vectorisation; the vector $\mathbf{w}$ with the zero vector. Thanks to the definition of $\mathcal{X}$, we know from Lemma 8 (and the composition of Lipschitz functions) that the loss is $\sqrt{2}\eta$-Lipschitz *w.r.t.* $\mathbf{w}$. For neural networks, we consider fully connected ReLU neural networks with $L$ hidden layers and $D$ nodes, where the leaky ReLU activation function $\text{ReLU}: \mathbb{R}^D \to \mathbb{R}^D$ applies elementwise $x \mapsto \max(x, 0.01x)$. More precisely, the network is defined by $h_{\mathbf{w}}(\mathbf{x}) = Wh^L(\cdots h^1(\mathbf{x})) + b$ where $W \in \mathbb{R}^{|\mathcal{Y}| \times D}, b \in \mathbb{R}^{|\mathcal{Y}|}$. Each layer $h^i(\mathbf{x}) = \text{ReLU}(W_i\mathbf{x} + b_i)$ has a weight matrix $W_i \in \mathbb{R}^{D \times D}$ and bias $b_i \in \mathbb{R}^D$ except for $i = 1$ where we have $W_1 \in \mathbb{R}^{D \times d}$. The weights $\mathbf{w}$ are also the vectorisation $\mathbf{w} = \text{vec}(\{W, W_L, \ldots, W_1, b, b_L, \ldots, b_1\})$. We have precised in Lemma 9 that our loss is Lipschitz *w.r.t.* the weights $\mathbf{w}$. We initialise the network similarly to [DR17] by sampling the weights from a Gaussian distribution with zero mean and a standard deviation of $\sigma = 0.04$; the weights are further clipped between $-2\sigma$ and $+2\sigma$. Moreover, the values in the biases $b_1, \ldots, b_L$ are set to 0.1, while the values for $b$ are set to 0. In the following, we consider $D = 600$ and $L = 2$; more experiments are considered in the appendix.

**Optimisation.** To perform the gradient steps, we use the COCOB-Backprop optimiser [OT17] (with parameter $\alpha = 10000$).[2] This optimiser is flexible as the learning rate is adaptive and, thus, does

---

[2]The parameter $\alpha$ in COCOB-Backprop can be seen as an initial learning rate; see [OT17].

not require hyperparameter tuning. For Algorithm 1, which solves Equation (5), we fix a batch size of 100, *i.e.*, $|\mathcal{U}| = 100$, and the number of epochs $T$ and $T'$ are fixed to perform at least 20000 iterations. Regarding Algorithm 2, which solves Equation (7), we set $t = 100$ for the log barrier, which is enough to constrain the weights and the number of iterations to $T = 10$.

**Datasets.** We study the performance of Algorithms 1 and 2 on UCI datasets [DG17] along with MNIST [LeC98] and FashionMNIST [XRV17]. We also split all the data (from the original training/test set) in two halves; the first part of the data serves in the algorithm (and is considered as a training set), while the second part is used to approximate the population risks $\mathfrak{R}_\mu(h)$ and $\mathfrak{C}_\mu$ (and considered as a testing set).

### 4.3   Results

We present in Table 1 the performance of Algorithms 1 and 2 compared to the Empirical Risk Minimisation (ERM) and the Online Gradient Descent (OGD) with the COCOB-Backprop optimiser. Tables 1a and 1c present the results of Algorithm 1 for the *i.i.d.* setting on linear and neural networks respectively, while Tables 1b and 1d present the results of Algorithm 2 for the online case.

**Analysis of the results.** In batch learning, we note that the regularisation term brings generalisation improvements compared to the empirical risk minimisation. Indeed, our batch algorithm (Algorithm 1) has a lower population risk $\mathfrak{R}_\mu(h)$ on 11 datasets for the linear models and 9 datasets for the neural networks. In particular, notice that NNs obtained from Algorithm 1 are more efficient than the ones obtained from ERM on MNIST and FASHIONMNIST, which are the more challenging datasets. This suggests that the regularisation term helps to generalise well. For the online case, the performance of the linear models obtained from our algorithm (Algorithm 2) and by OGD are comparable: we have a tighter population classification risk $\mathfrak{R}_\mu(h)$ on 5 datasets over 13. However, notice that the risk difference is less than 0.05 on 6 datasets. The advantage of Algorithm 2 is more pronounced for neural networks: we improve the performance in all datasets except ADULT and SENSORLESS. Hence, this confirms that optimising the regularised loss $\ell(h_\mathbf{w}, \mathbf{z}_i) + \|\mathbf{w} - \mathbf{w}_{i-1}\|$ brings a good advantage compared to the loss $\ell(h_\mathbf{w}, \mathbf{z}_i)$ only. A possible explanation would be that OGD suffers from underfitting (with a high empirical risk $\mathfrak{C}_\mu$) while we are able to control overfitting through a regularisation term. Indeed, only one gradient descent step is done for each new datum $(\mathbf{x}_i, y_i)$, which might not be sufficient to decrease the loss. Instead, our method solves the problem associated with Equation (7) and constrains the descent with the norm $\|\mathbf{w} - \mathbf{w}_{i-1}\|$.

## 5   Conclusion and Perspectives

We derived novel generalisation bounds based on the Wasserstein distance, both for batch and online learning, allowing for the use of deterministic hypotheses through PAC-Bayes. We derived new learning algorithms which are inspired by the bounds, with remarkable empirical performance on a number of datasets: we hope our work can pave the way to promising future developments (both theoretical and practical) of generalisation bounds based on the Wasserstein distance. Given the mostly theoretical nature of our work, we do not foresee an immediate negative societal impact, although we hope a better theoretical understanding of generalisation will ultimately benefit practitioners of machine learning algorithms and encourage virtuous initiatives.

## 6   Acknowledgements

We warmly thank the reviewers who provided insightful comments and suggestions which greatly helped us improve our manuscript. P.V. and U.S. are partially supported by the French government under the management of Agence Nationale de la Recherche as part of the "Investissements d'avenir" program, reference ANR-19-P3IA-0001 (PRAIRIE 3IA Institute). U.S. is also partially supported by the European Research Council Starting Grant DYNASTY – 101039676. B.G. acknowledges partial support by the U.S. Army Research Laboratory and the U.S. Army Research Office, and by the U.K. Ministry of Defence and the U.K. Engineering and Physical Sciences Research Council (EPSRC) under grant number EP/R013616/1. B.G. acknowledges partial support from the French National Agency for Research, grants ANR-18-CE40-0016-01 and ANR-18-CE23-0015-02.

Table 1: Performance of Algorithms 1 and 2 compared respectively to ERM and OGD on different datasets on linear and neural network models. For the *i.i.d.* setting, we consider $\varepsilon = 1/m$ and $\varepsilon = 1/\sqrt{m}$ and with $K = 0.2\sqrt{m}$. For each method, we plot the empirical risk $\mathfrak{R}_{\mathcal{S}}(h)$ or $\mathfrak{C}_{\mathcal{S}}$ with its associated test risk $\mathfrak{R}_{\mu}(h)$ or $\mathfrak{C}_{\mu}$. The risk in **bold** corresponds to the lowest one among the ones considered. For the online case, the two population risks are underlined when the absolute difference is lower than 0.05.

(a) Linear model – batch learning

| Dataset | Alg. 1 ($\frac{1}{m}$) $\mathfrak{R}_{\mathcal{S}}(h)$ | $\mathfrak{R}_{\mu}(h)$ | Alg. 1 ($\frac{1}{\sqrt{m}}$) $\mathfrak{R}_{\mathcal{S}}(h)$ | $\mathfrak{R}_{\mu}(h)$ | ERM $\mathfrak{R}_{\mathcal{S}}(h)$ | $\mathfrak{R}_{\mu}(h)$ |
|---|---|---|---|---|---|---|
| ADULT | .165 | **.166** | .165 | .167 | .166 | .167 |
| FASHIONMNIST | .128 | .151 | .126 | **.148** | .139 | .153 |
| LETTER | .285 | .297 | .287 | **.296** | .287 | .297 |
| MNIST | .200 | .216 | .066 | .092 | .065 | **.091** |
| MUSHROOMS | .001 | **.001** | .001 | **.001** | .001 | **.001** |
| NURSERY | .766 | **.773** | .760 | **.773** | .794 | .807 |
| PENDIGITS | .049 | **.059** | .050 | .061 | .052 | .064 |
| PHISHING | .063 | **.067** | .065 | .069 | .064 | **.067** |
| SATIMAGE | .144 | **.200** | .138 | .201 | .148 | .209 |
| SEGMENTATION | .057 | **.216** | .164 | .386 | .087 | .232 |
| SENSORLESS | .129 | **.129** | .131 | .131 | .134 | .136 |
| TICTACTOE | .388 | .299 | .013 | **.021** | .228 | .238 |
| YEAST | .527 | .497 | .524 | .504 | .470 | **.427** |

(b) Linear model – online learning

| Dataset | Alg. 2 $\mathfrak{C}_{\mathcal{S}}$ | $\mathfrak{C}_{\mu}$ | OGD $\mathfrak{C}_{\mathcal{S}}$ | $\mathfrak{C}_{\mu}$ |
|---|---|---|---|---|
| ADULT | .230 | **.236** | .248 | .248 |
| FASHIONMNIST | .223 | **.282** | .540 | .548 |
| LETTER | .919 | .935 | .916 | **.926** |
| MNIST | .284 | **.310** | .378 | .397 |
| MUSHROOMS | .218 | .222 | .082 | **.087** |
| NURSERY | .794 | .807 | .789 | **.805** |
| PENDIGITS | .342 | **.484** | .589 | .600 |
| PHISHING | .226 | .242 | .226 | **.220** |
| SATIMAGE | .669 | .938 | .635 | **.888** |
| SEGMENTATION | .749 | **.803** | .738 | .893 |
| SENSORLESS | .906 | .910 | .825 | **.830** |
| TICTACTOE | .443 | .468 | .390 | **.303** |
| YEAST | .699 | .713 | .667 | **.708** |

(c) NN model – batch learning

| Dataset | Alg. 1 ($\frac{1}{m}$) $\mathfrak{R}_{\mathcal{S}}(h)$ | $\mathfrak{R}_{\mu}(h)$ | Alg. 1 ($\frac{1}{\sqrt{m}}$) $\mathfrak{R}_{\mathcal{S}}(h)$ | $\mathfrak{R}_{\mu}(h)$ | ERM $\mathfrak{R}_{\mathcal{S}}(h)$ | $\mathfrak{R}_{\mu}(h)$ |
|---|---|---|---|---|---|---|
| ADULT | .164 | .164 | .166 | .165 | .165 | **.163** |
| FASHIONMNIST | .159 | .163 | .156 | **.160** | .163 | .167 |
| LETTER | .259 | .272 | .250 | **.260** | .258 | .270 |
| MNIST | .112 | .120 | .084 | **.094** | .119 | .127 |
| MUSHROOMS | .000 | **.000** | .000 | **.000** | .000 | **.000** |
| NURSERY | .706 | **.719** | .706 | **.719** | .706 | **.719** |
| PENDIGITS | .009 | .023 | .021 | .032 | .009 | **.022** |
| PHISHING | .042 | **.050** | .039 | .054 | .046 | .055 |
| SATIMAGE | .132 | .184 | .149 | **.172** | .141 | .189 |
| SEGMENTATION | .145 | **.250** | .189 | .373 | .174 | .389 |
| SENSORLESS | .076 | .079 | .077 | .079 | .075 | **.078** |
| TICTACTOE | .392 | .301 | .000 | .038 | .000 | **.023** |
| YEAST | .679 | .666 | .487 | **.478** | .644 | .682 |

(d) NN model – online learning

| Dataset | Alg. 2 $\mathfrak{C}_{\mathcal{S}}$ | $\mathfrak{C}_{\mu}$ | OGD $\mathfrak{C}_{\mathcal{S}}$ | $\mathfrak{C}_{\mu}$ |
|---|---|---|---|---|
| ADULT | .241 | .254 | .248 | **.248** |
| FASHIONMNIST | .096 | **.327** | .397 | .446 |
| LETTER | .829 | .945 | .958 | .963 |
| MNIST | .092 | **.265** | .470 | .521 |
| MUSHROOMS | .082 | **.122** | .202 | .217 |
| NURSERY | .800 | **.805** | .793 | .806 |
| PENDIGITS | .323 | **.537** | .871 | .879 |
| PHISHING | .164 | **.222** | .331 | .318 |
| SATIMAGE | .401 | **.763** | .626 | .857 |
| SEGMENTATION | .619 | **.857** | .739 | .913 |
| SENSORLESS | .899 | .910 | .622 | **.633** |
| TICTACTOE | .388 | **.309** | .397 | **.309** |
| YEAST | .662 | **.720** | .702 | **.720** |

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

The supplementary material is organized as follows:

*(i)* We provide more discussion about Theorems 1 and 2 in Appendix A;

*(ii)* The proofs of Theorems 1 to 4 are presented in Appendix B;

*(iii)* We present in Appendix C additional information about the experiments.

# A  Additional insights on Section 3.1

In Appendix A.1, we provide additional discussion about Theorem 1 while Appendix A.2 discuss about the convergence rates for Theorem 2.

## A.1  Supplementary discussion about Theorem 1

[HG23b, Corollary 10] proposed PAC-Bayes bounds with Wasserstein distances on a Euclidean predictor space with Gaussian prior and posteriors. The bounds have an explicit convergence rate of $\mathcal{O}(\sqrt{dW_1(\rho,\pi)/m})$ where the predictor space is Euclidean with dimension $d$. While our bound does not propose such an explicit convergence rate, it allows us to derive learning algorithms as described in Section 4. A broader discussion about the role of $K$ is detailed in Theorem 2. Furthermore, our bound holds for any Polish predictor space and does not require Gaussian distributions. Furthermore, our result exploits data-dependent priors and deals with the dimension only through the Wasserstein distance, which can attenuate the impact of the dimension.

## A.2  Convergence rates for Theorem 2

In this section, we discuss more deeply the values of $K$ in Theorem 2. This implies a tradeoff between the number of sets $K$ and the cardinal of each $\mathcal{S}_i$. The tightness of the bound depends highly on the sets $\mathcal{S}_1,\ldots,\mathcal{S}_K$.

**Full batch setting K=1.** When $\mathcal{S}_1 = \mathcal{S}$ with $K = 1$, the bound of Theorem 2 becomes, with probability $1-\delta$, for any $\rho \in \mathcal{M}(\mathcal{H})$

$$\mathop{\mathbb{E}}_{h\sim\rho}\left[\mathrm{R}_\mu(h) - \hat{\mathrm{R}}_{\mathcal{S}}(h)\right] \leq 2L\mathrm{W}(\rho,\pi) + 2\sqrt{\frac{\ln\frac{1}{\delta}}{m}}\,,$$

where $\pi = \pi_1$ is data-free. This bound can be seen as the high-probability (PAC-Bayesian) version of the expected bound of [WDFC19]. Furthermore, in this setting, we are able, through our proof technique, to recover an explicit convergence rate similar to the one of [AEMM22, Theorem 12]. It is stated below.

**Corollary 5.** *For any distribution $\mu$ on $\mathcal{Z}$, for any finite hypothesis space $\mathcal{H}$ equipped with a distance $d$, for any $L$-Lipschitz loss function $\ell : \mathcal{H} \times \mathcal{Z} \to [0,1]$, for any $\delta \in (0,1]$, we have, with probability $1-\delta$ over the sample $\mathcal{S}$, for any $\rho \in \mathcal{M}(\mathcal{H})$:*

$$\mathop{\mathbb{E}}_{h\sim\rho}\left[R_\mu(h) - \hat{R}_{\mathcal{S}}(h)\right] \leq L\sqrt{\frac{2\ln\left(4|\mathcal{H}|^2/\delta\right)}{m}}\mathrm{W}(\rho,\pi) + 2\sqrt{\frac{\ln\left(2/\delta\right)}{m}}$$

*where $\pi$ is a data-free prior.*

*Proof.* We exploit [AEMM22, Equation 35] to state that with probability at least $1 - \delta/2$, for any $(h,h') \in \mathcal{H}^2$:

$$\left|\frac{1}{m}\sum_{i=1}^{m}\left[\ell\left(h',\mathbf{z}_i\right) - \ell\left(h,\mathbf{z}_i\right)\right] - \mathop{\mathbb{E}}_{\mathbf{z}\sim\mu}\left[\ell\left(h',\mathbf{z}\right) - \ell(h,\mathbf{z})\right]\right| \leq L\sqrt{\frac{2\ln\left(\frac{4|\mathcal{H}|^2}{\delta}\right)}{m}}d\left(h,h'\right).$$

So, with high probability, we can exploit the Kantorovich-Rubinstein duality with this new Lipschitz constant: with probability at least $1 - \delta/2$:

$$\mathop{\mathbb{E}}_{h\sim\rho}\left[\mathrm{R}_\mu(h) - \hat{\mathrm{R}}_{\mathcal{S}}(h)\right] \leq L\sqrt{\frac{2\ln\left(\frac{4|\mathcal{H}|^2}{\delta}\right)}{m}}\mathrm{W}(\rho,\pi) + \mathop{\mathbb{E}}_{h\sim\pi}\frac{1}{m}\left[\sum_{i=1}^{m}\mathrm{R}_\mu(h) - \ell(h,\mathbf{z}_i)\right],$$

To conclude, we control the quantity on the right-hand side the same way as in Theorem 1 and Theorem 2. We then have, with probability at least $1 - \delta/2$, for a loss function in $[0,1]$:

$$\frac{1}{m}\sum_{i=1}^{m} R_\mu(h) - \ell(h, \mathbf{z}_i) \le 2\sqrt{\frac{\ln \frac{K}{\delta}}{m}}.$$

Taking the union bound concludes the proof. $\qquad\qquad\qquad\qquad\qquad\qquad\qquad\qquad\square$

**Mini-batch setting $K = \sqrt{m}$.** When a tradeoff is desired between the quantity of data we want to infuse in our priors and an explicit convergence rate, a meaningful candidate is when $K = \sqrt{m}$. Theorem 2's bound becomes, in this particular case:

$$\mathbb{E}_{h \sim \rho}\left[R_\mu(h) - \hat{R}_\mathcal{S}(h)\right] \le \frac{2L}{\sqrt{m}}\sum_{i=1}^{\sqrt{m}} W(\rho, \pi_i) + 2\sqrt{\frac{\ln \frac{\sqrt{m}}{\delta}}{\sqrt{m}}}. \tag{8}$$

**Towards online learning: $K = m$.** When $K = m$, the sets $\mathcal{S}_i$ contain only one example. More precisely, we have for all $i \in \{1, \ldots, m\}$ the set $\mathcal{S}_i = \{\mathbf{z}_i\}$. In this case, the bound becomes:

$$\mathbb{E}_{h \sim \rho}\left[R_\mu(h) - \hat{R}_\mathcal{S}(h)\right] \le \frac{2L}{m}\sum_{i=1}^{m} W(\rho, \pi_i) + 2\sqrt{\ln \frac{m}{\delta}}.$$

This bound is vacuous since the last term is incompressible, hence the need for a new technique detailed in Section 3.2 to deal with it.

# B   Proofs

The proof of Theorem 1 is presented in Appendix B.1. Appendices B.2 and B.3 introduce two proofs of Theorem 2. Theorem 3's proof is presented in Appendix B.4. Appendix B.5 provides the proof of Theorem 3.

## B.1   Proof of Theorem 1

**Theorem 1.** *We assume the loss $\ell$ to be $L$-Lipschitz. Then, for any $\delta \in (0,1]$, for any sequence of positive scalar $(\lambda_i)_{i \in \{1, \ldots, K\}}$, with probability at least $1 - \delta$ over the sample $\mathcal{S}$, the following holds for the distributions $\pi_{i,\mathcal{S}} := \pi_i(\mathcal{S}, .)$ and for any $\rho \in \mathcal{M}(\mathcal{H})$:*

$$\mathbb{E}_{h \sim \rho}\left[R_\mu(h) - \hat{R}_\mathcal{S}(h)\right]$$
$$\le \sum_{i=1}^{K} \frac{2|\mathcal{S}_i|L}{m} W(\rho, \pi_{i,\mathcal{S}}) + \frac{1}{m}\sum_{i=1}^{K} \frac{\ln\left(\frac{K}{\delta}\right)}{\lambda_i} + \frac{\lambda_i}{2}\left(\mathbb{E}_{h \sim \pi_{i,\mathcal{S}}}\left[\hat{V}_{|\mathcal{S}_i|}(h) + V_{|\mathcal{S}_i|}(h)\right]\right),$$

*where $\pi_{i,\mathcal{S}}$ does not depend on $\mathcal{S}_i$. Also, for any $i, |S_i|$, we have $\hat{V}_{|\mathcal{S}_i|}(h) = \sum_{\mathbf{z} \in \mathcal{S}_i} (\ell(h, \mathbf{z}) - R_\mu(h))^2$ and $V_{|\mathcal{S}_i|}(h) = \mathbb{E}_{\mathcal{S}_i}\left[\hat{V}_{|\mathcal{S}_i|}(h)\right]$.*

*Proof.* For the sake of readability, we identify, for any $i$, $\pi_i$ and $\pi_{i,\mathcal{S}}$.

**Step 1: Exploit the Kantorovich duality [Vil09, Remark 6.5].**   First of all, note that for a $L$-Lipschitz loss function $\ell : \mathcal{H} \times \mathcal{Z} \to [0,1]$, we have

$$\left|\left(|\mathcal{S}_i|R_\mu(h_1) - \sum_{\mathbf{z} \in \mathcal{S}_i} \ell(h_1, \mathbf{z})\right) - \left(|\mathcal{S}_i|R_\mu(h_2) - \sum_{\mathbf{z} \in \mathcal{S}_i} \ell(h_2, \mathbf{z})\right)\right| \le 2|\mathcal{S}_i|Ld(h_1, h_2). \tag{9}$$

Indeed, we can deduce Equation (9) from Jensen inequality, the triangle inequality, and by definition that we have

$$
\left| \left( |\mathcal{S}_i| \mathrm{R}_\mu(h_1) - \sum_{\mathbf{z} \in \mathcal{S}_i} \ell(h_1, \mathbf{z}) \right) - \left( |\mathcal{S}_i| \mathrm{R}_\mu(h_2) - \sum_{\mathbf{z} \in \mathcal{S}_i} \ell(h_2, \mathbf{z}) \right) \right|
$$

$$
= \left| \left( \sum_{\mathbf{z} \in \mathcal{S}_i} \mathrm{R}_\mu(h_1) - \sum_{\mathbf{z} \in \mathcal{S}_i} \ell(h_1, \mathbf{z}) \right) - \left( \sum_{\mathbf{z} \in \mathcal{S}_i} \mathrm{R}_\mu(h_2) - \sum_{\mathbf{z} \in \mathcal{S}_i} \ell(h_2, \mathbf{z}) \right) \right|
$$

$$
\leq \sum_{\mathbf{z} \in \mathcal{S}_i} \mathop{\mathbb{E}}_{\mathbf{z}' \sim \mu} \left[ |\ell(h_1, \mathbf{z}') - \ell(h_2, \mathbf{z}')| + |\ell(h_2, \mathbf{z}) - \ell(h_1, \mathbf{z})| \right]
$$

$$
\leq \mathop{\mathbb{E}}_{\mathbf{z}' \sim \mu} \sum_{\mathbf{z} \in \mathcal{S}_i} 2L d(h_1, h_2)
$$

$$
= 2|\mathcal{S}_i| L d(h_1, h_2).
$$

We are now able to upper-bound $\mathbb{E}_{h \sim \rho}[\mathrm{R}_\mu(h) - \hat{\mathrm{R}}_\mathcal{S}(h)]$. Indeed, we have

$$
\mathop{\mathbb{E}}_{h \sim \rho} \left[ \mathrm{R}_\mu(h) - \hat{\mathrm{R}}_\mathcal{S}(h) \right] = \frac{1}{m} \sum_{i=1}^{K} \mathop{\mathbb{E}}_{h \sim \rho} \left[ |\mathcal{S}_i| \mathrm{R}_\mu(h) - \sum_{\mathbf{z} \in \mathcal{S}_i} \ell(h, \mathbf{z}) \right]
$$

$$
\leq \sum_{i=1}^{K} \frac{2|\mathcal{S}_i| L}{m} \mathrm{W}(\rho, \pi_i) + \sum_{i=1}^{K} \mathop{\mathbb{E}}_{h \sim \pi_i} \frac{1}{m} \left[ |\mathcal{S}_i| \mathrm{R}_\mu(h) - \sum_{\mathbf{z} \in \mathcal{S}_i} \ell(h, \mathbf{z}) \right], \quad (10)
$$

where the inequality comes from the Kantorovich-Rubinstein duality theorem.

**Step 2: Define an adapted supermartingale.** For any $1 \leq i \leq K$, we fix $\lambda_i > 0$ and we provide an arbitrary order to the elements of $\mathcal{S}_i := \{\mathbf{z}_{i,1}, \cdots, \mathbf{z}_{i,|\mathcal{S}_i|}\}$. Then we define for any $h$:

$$
M_{|\mathcal{S}_i|}(h) := |\mathcal{S}_i| \mathrm{R}_\mu(h) - \sum_{\mathbf{z} \in \mathcal{S}_i} \ell(h, \mathbf{z}) = \sum_{j=1}^{|\mathcal{S}_i|} R_\mu(h) - \ell(h, \mathbf{z}_{i,j}).
$$

Remark that, because our data are *i.i.d.*, $(M_{|\mathcal{S}_i|})_{|\mathcal{S}_i| \geq 1}$ is a martingale. We then exploit the technique [HG23a] to define a supermartingale. More precisely, we exploit a result from [BT08] cited in Lemma 1.3 of [HG23a] coupled with Lemma 2.2 of [HG23a] to ensure that the process

$$
SM_{|\mathcal{S}_i|} := \mathop{\mathbb{E}}_{h \sim \pi_i} \left[ \exp \left( \lambda_i M_{|\mathcal{S}_i|}(h) - \frac{\lambda_i^2}{2} \left( \hat{V}_{|\mathcal{S}_i|}(h) + V_{|\mathcal{S}_i|}(h) \right) \right) \right],
$$

is a supermartingale, where $\hat{V}_{|\mathcal{S}_i|}(h) = \sum_{j=1}^{|\mathcal{S}_i|} (\ell(h, \mathbf{z}_{i,j}) - R_\mu(h))^2$ and $V_{|\mathcal{S}_i|}(h) = \mathbb{E}_{\mathcal{S}_i} \left[ \hat{V}_{|\mathcal{S}_i|}(h) \right]$.

**Step 3. Combine steps 1 and 2.** We restart from Equation (10) to exploit again the Kantorovich-Rubinstein duality.

$$
\mathop{\mathbb{E}}_{h \sim \rho} \left[ \mathrm{R}_\mu(h) - \hat{\mathrm{R}}_\mathcal{S}(h) \right] = \frac{1}{m} \sum_{i=1}^{K} \mathop{\mathbb{E}}_{h \sim \rho} \left[ |\mathcal{S}_i| \mathrm{R}_\mu(h) - \sum_{\mathbf{z} \in \mathcal{S}_i} \ell(h, \mathbf{z}) \right]
$$

$$
\leq \sum_{i=1}^{K} \frac{2|\mathcal{S}_i| L}{m} \mathrm{W}(\rho, \pi_i) + \sum_{i=1}^{K} \frac{1}{m \lambda_i} \lambda_i \mathop{\mathbb{E}}_{h \sim \pi_i} \left[ |\mathcal{S}_i| \mathrm{R}_\mu(h) - \sum_{\mathbf{z} \in \mathcal{S}_i} \ell(h, \mathbf{z}) \right],
$$

$$
= \sum_{i=1}^{K} \frac{2|\mathcal{S}_i| L}{m} \mathrm{W}(\rho, \pi_i) + \sum_{i=1}^{K} \frac{1}{m \lambda_i} \mathop{\mathbb{E}}_{h \sim \pi_i} \left[ \lambda_i M_{|\mathcal{S}_i|} \right],
$$

$$
\leq \sum_{i=1}^{K} \frac{2|\mathcal{S}_i| L}{m} \mathrm{W}(\rho, \pi_i) + \sum_{i=1}^{K} \frac{1}{m \lambda_i} \ln \left( SM_{|\mathcal{S}_i|} \right)
$$

$$
+ \frac{1}{m} \sum_{i=1}^{K} \mathop{\mathbb{E}}_{h \sim \pi_i} \left[ \frac{\lambda_i}{2} \left( \hat{V}_{|\mathcal{S}_i|}(h) + V_{|\mathcal{S}_i|}(h) \right) \right].
$$

The last line holds thanks to Jensen's inequality. We now apply Ville's inequality (see *e.g.*, Section 1.2 of [HG23a]). We have for any $i$:

$$\mathbb{P}_{\mathcal{S}_i \sim \mu^{|\mathcal{S}_i|}} \left( \forall |S_i| \geq 1, SM_{|S_i|} \leq \frac{1}{\delta} \right) \geq 1 - \delta.$$

Applying an union bound and authorising $\lambda_i$ to be a function of $|S_i|$ (thus the inequality does not hold for all $|\mathcal{S}_i|$ simultaneously) finally gives with probability at least $1 - \delta$, for all $\rho \in \mathcal{M}(\mathcal{H})$ :

$$\mathbb{E}_{h \sim \rho} \left[ \mathrm{R}_\mu(h) - \hat{\mathrm{R}}_{\mathcal{S}}(h) \right] \leq \sum_{i=1}^K \frac{2|\mathcal{S}_i|L}{m} \mathrm{W}(\rho, \pi_i) + \sum_{i=1}^K \frac{\ln\left(\frac{K}{\delta}\right)}{\lambda_i m} + \frac{\lambda_i}{2m} \mathbb{E}_{h \sim \pi_i} \left[ \hat{V}_{|\mathcal{S}_i|}(h) + V_{|\mathcal{S}_i|}(h) \right].$$

$\square$

## B.2  Proof of Theorem 2

**Theorem 2.** *We assume our loss $\ell$ to be non-negative and $L$-Lipschitz. We also assume that, for any $1 \leq i \leq K$, for any dataset$\mathcal{S}$, we have $\mathbb{E}_{h \sim \pi_i(.,\mathcal{S}), z \sim \mu} \left[ \ell(h, z)^2 \right] \leq 1$ (bounded order 2 moments for priors). Then, for any $\delta \in (0, 1]$, with probability at least $1 - \delta$ over the sample $\mathcal{S}$, the following holds for the distributions $\pi_{i,\mathcal{S}} := \pi_i(\mathcal{S}, .)$ and for any $\rho \in \mathcal{M}(\mathcal{H})$:*

$$\mathbb{E}_{h \sim \rho} \left[ R_\mu(h) - \hat{R}_{\mathcal{S}}(h) \right] \leq \sum_{i=1}^K \frac{2|\mathcal{S}_i|L}{m} \mathrm{W}(\rho, \pi_{i,\mathcal{S}}) + \sum_{i=1}^K \sqrt{\frac{2|\mathcal{S}_i| \ln \frac{K}{\delta}}{m^2}},$$

*where $\pi_{i,\mathcal{S}}$ does not depend on $\mathcal{S}_i$.*

*Proof.* For the sake of readability, we identify, for any $i$, $\pi_i$ and $\pi_{i,\mathcal{S}}$.

**Step 1: Exploit the Kantorovich duality [Vil09, Remark 6.5].**  First of all, note that for a $L$-Lipschitz loss function $\ell : \mathcal{H} \times \mathcal{Z} \to [0, 1]$, we have

$$\left| \left( |\mathcal{S}_i| \mathrm{R}_\mu(h_1) - \sum_{\mathbf{z} \in \mathcal{S}_i} \ell(h_1, \mathbf{z}) \right) - \left( |\mathcal{S}_i| \mathrm{R}_\mu(h_2) - \sum_{\mathbf{z} \in \mathcal{S}_i} \ell(h_2, \mathbf{z}) \right) \right| \leq 2|\mathcal{S}_i| L d(h_1, h_2). \tag{11}$$

Indeed, we can deduce Equation (11) from Jensen inequality, the triangle inequality, and by definition that we have

$$\left| \left( |\mathcal{S}_i| \mathrm{R}_\mu(h_1) - \sum_{\mathbf{z} \in \mathcal{S}_i} \ell(h_1, \mathbf{z}) \right) - \left( |\mathcal{S}_i| \mathrm{R}_\mu(h_2) - \sum_{\mathbf{z} \in \mathcal{S}_i} \ell(h_2, \mathbf{z}) \right) \right|$$

$$= \left| \left( \sum_{\mathbf{z} \in \mathcal{S}_i} \mathrm{R}_\mu(h_1) - \sum_{\mathbf{z} \in \mathcal{S}_i} \ell(h_1, \mathbf{z}) \right) - \left( \sum_{\mathbf{z} \in \mathcal{S}_i} \mathrm{R}_\mu(h_2) - \sum_{\mathbf{z} \in \mathcal{S}_i} \ell(h_2, \mathbf{z}) \right) \right|$$

$$\leq \sum_{\mathbf{z} \in \mathcal{S}_i} \mathbb{E}_{\mathbf{z}' \sim \mu} \left[ |\ell(h_1, \mathbf{z}') - \ell(h_2, \mathbf{z}')| + |\ell(h_2, \mathbf{z}) - \ell(h_1, \mathbf{z})| \right]$$

$$\leq \mathbb{E}_{\mathbf{z}' \sim \mu} \sum_{\mathbf{z} \in \mathcal{S}_i} 2L d(h_1, h_2)$$

$$= 2|\mathcal{S}_i| L d(h_1, h_2).$$

We are now able to upper-bound $\mathbb{E}_{h \sim \rho}[\mathrm{R}_\mu(h) - \hat{\mathrm{R}}_{\mathcal{S}}(h)]$. Indeed, we have

$$\mathbb{E}_{h \sim \rho} \left[ \mathrm{R}_\mu(h) - \hat{\mathrm{R}}_{\mathcal{S}}(h) \right] = \frac{1}{m} \sum_{i=1}^K \mathbb{E}_{h \sim \rho} \left[ |\mathcal{S}_i| \mathrm{R}_\mu(h) - \sum_{\mathbf{z} \in \mathcal{S}_i} \ell(h, \mathbf{z}) \right]$$

$$\leq \sum_{i=1}^K \frac{2|\mathcal{S}_i|L}{m} \mathrm{W}(\rho, \pi_i) + \sum_{i=1}^K \mathbb{E}_{h \sim \pi_i} \frac{1}{m} \left[ |\mathcal{S}_i| \mathrm{R}_\mu(h) - \sum_{\mathbf{z} \in \mathcal{S}_i} \ell(h, \mathbf{z}) \right], \tag{12}$$

where the inequality comes from the Kantorovich-Rubinstein duality theorem.

**Step 2: Define an adapted supermartingale.** For any $1 \leq i \leq K$, we fix $\lambda_i > 0$ and we provide an arbitrary order to the elements of $\mathcal{S}_i := \{\mathbf{z}_{i,1}, \cdots, \mathbf{z}_{i,|\mathcal{S}_i|}\}$. Then we define for any $h$:

$$M_{|\mathcal{S}_i|}(h) := |\mathcal{S}_i| \mathbf{R}_\mu(h) - \sum_{\mathbf{z} \in \mathcal{S}_i} \ell(h, \mathbf{z}) = \sum_{j=1}^{|\mathcal{S}_i|} R_\mu(h) - \ell(h, \mathbf{z}_{i,j}).$$

Remark that, because our data are *i.i.d.*, $(M_{|\mathcal{S}_i|})_{|\mathcal{S}_i| \geq 1}$ is a martingale. We then exploit the technique [CWR23] to define a supermartingale. More precisely, we exploit [CWR23, Lemma A.2 and Lemma B.1] to ensure that the process

$$SM_{|\mathcal{S}_i|} := \mathbb{E}_{h \sim \pi_i} \left[ \exp \left( \lambda_i M_{|\mathcal{S}_i|}(h) - \frac{\lambda_i^2}{2} L_{|\mathcal{S}_i|}(h) \right) \right],$$

is a supermartingale, where, because our data are *i.i.d.*, $L_{|\mathcal{S}_i|}(h) = \mathbb{E}_{\mathcal{S}} \left[ \sum_{j=1}^{|S_i|} \ell(h, \mathbf{z}_{i,j})^2 \right] = |\mathcal{S}_i| \mathbb{E}_{z \sim \mu}[\ell(h, z)^2]$.

**Step 3. Combine steps 1 and 2.** We restart from Equation (12) to exploit the Kantorovich-Rubinstein duality again.

$$
\begin{aligned}
\mathbb{E}_{h \sim \rho} \left[ \mathbf{R}_\mu(h) - \hat{\mathbf{R}}_\mathcal{S}(h) \right] &= \frac{1}{m} \sum_{i=1}^{K} \mathbb{E}_{h \sim \rho} \left[ |\mathcal{S}_i| \mathbf{R}_\mu(h) - \sum_{\mathbf{z} \in \mathcal{S}_i} \ell(h, \mathbf{z}) \right] \\
&\leq \sum_{i=1}^{K} \frac{2|\mathcal{S}_i| L}{m} \mathrm{W}(\rho, \pi_i) + \sum_{i=1}^{K} \frac{1}{m\lambda_i} \lambda_i \mathbb{E}_{h \sim \pi_i} \left[ |\mathcal{S}_i| \mathbf{R}_\mu(h) - \sum_{\mathbf{z} \in \mathcal{S}_i} \ell(h, \mathbf{z}) \right], \\
&= \sum_{i=1}^{K} \frac{2|\mathcal{S}_i| L}{m} \mathrm{W}(\rho, \pi_i) + \sum_{i=1}^{K} \frac{1}{m\lambda_i} \mathbb{E}_{h \sim \pi_i} \left[ \lambda_i M_{|\mathcal{S}_i|} \right], \\
&\leq \sum_{i=1}^{K} \frac{2|\mathcal{S}_i| L}{m} \mathrm{W}(\rho, \pi_i) + \sum_{i=1}^{K} \frac{1}{m\lambda_i} \ln \left( SM_{|\mathcal{S}_i|} \right) \\
&\quad + \frac{1}{m} \sum_{i=1}^{K} \mathbb{E}_{h \sim \pi_i} \left[ \frac{\lambda_i}{2} L_{|\mathcal{S}_i|}(h) \right].
\end{aligned}
$$

The last line holds thanks to Jensen's inequality. We now apply Ville's inequality (see *e.g.*, section 1.2 of [HG23a]). We have for any $i$:

$$\mathbb{P}_{\mathcal{S}_i \sim \mu^{|\mathcal{S}_i|}} \left( \forall |S_i| \geq 1, SM_{|\mathcal{S}_i|} \leq \frac{1}{\delta} \right) \geq 1 - \delta.$$

Applying an union bound and authorising $\lambda_i$ to be a function of $|S_i|$ (thus the inequality does not hold for all $|\mathcal{S}_i|$ simultaneously) finally gives with probability at least $1 - \delta$, for all $\rho \in \mathcal{M}(\mathcal{H})$ :

$$\mathbb{E}_{h \sim \rho} \left[ \mathbf{R}_\mu(h) - \hat{\mathbf{R}}_\mathcal{S}(h) \right] \leq \sum_{i=1}^{K} \frac{2|\mathcal{S}_i| L}{m} \mathrm{W}(\rho, \pi_i) + \sum_{i=1}^{K} \frac{\ln \left( \frac{K}{\delta} \right)}{\lambda_i m} + \frac{\lambda_i}{2m} \mathbb{E}_{h \sim \pi_i} \left[ L |\mathcal{S}_i|(h) \right].$$

Finally, using the assumption $\mathbb{E}_{h \sim \pi_i} \mathbb{E}_{z \sim \mu}[\ell(h, z)^2] \leq 1$ gives, with probability at least $1 - \delta$, for all $\rho \in \mathcal{M}(\mathcal{H})$:

$$\mathbb{E}_{h \sim \rho} \left[ \mathbf{R}_\mu(h) - \hat{\mathbf{R}}_\mathcal{S}(h) \right] \leq \sum_{i=1}^{K} \frac{2|\mathcal{S}_i| L}{m} \mathrm{W}(\rho, \pi_i) + \sum_{i=1}^{K} \frac{\ln \left( \frac{K}{\delta} \right)}{\lambda_i m} + \frac{\lambda_i [\mathcal{S}_i|]}{2m}.$$

Taking for each $i$, $\lambda_i = \sqrt{\frac{2 \ln(K/\delta)}{|\mathcal{S}_i|}}$ concludes the proof. $\qquad\square$

### B.3 Alternative proof of Theorem 2

We state here a slightly tighter version of Theorem 2 for bounded losses, which relies on an application of McDiarmid's inequality instead of supermartingale techniques. This is useful for the numerical evaluations of our bound.

**Theorem 6.** *We assume our loss $\ell$ to be in $[0, 1]$ and $L$-Lipschitz. Then, for any $\delta \in (0, 1]$, with probability at least $1 - \delta$ over the sample $\mathcal{S}$, the following holds for the distributions $\pi_{i,\mathcal{S}} := \pi_i(\mathcal{S}, .)$ and for any $\rho \in \mathcal{M}(\mathcal{H})$:*

$$\mathop{\mathbb{E}}_{h \sim \rho} \left[ R_\mu(h) - \hat{R}_\mathcal{S}(h) \right] \leq \sum_{i=1}^{K} \frac{2|\mathcal{S}_i|L}{m} W(\rho, \pi_{i,\mathcal{S}}) + \sum_{i=1}^{K} \sqrt{\frac{|\mathcal{S}_i| \ln \frac{K}{\delta}}{2m^2}}$$

*where $\pi_i$ does not depend on $\mathcal{S}_i$.*

*Proof.* For the sake of readability, we identify, for any $i$, $\pi_i$ and $\pi_{i,\mathcal{S}}$.

First of all, note that for a $L$-Lipschitz loss function $\ell : \mathcal{H} \times \mathcal{Z} \to [0, 1]$, we have

$$\left| \left( |\mathcal{S}_i| R_\mu(h_1) - \sum_{\mathbf{z} \in \mathcal{S}_i} \ell(h_1, \mathbf{z}) \right) - \left( |\mathcal{S}_i| R_\mu(h_2) - \sum_{\mathbf{z} \in \mathcal{S}_i} \ell(h_2, \mathbf{z}) \right) \right| \leq 2|\mathcal{S}_i| L d(h_1, h_2). \tag{13}$$

Indeed, we can deduce Equation (13) from Jensen's inequality, the triangle inequality, and by definition that we have

$$\left| \left( |\mathcal{S}_i| R_\mu(h_1) - \sum_{\mathbf{z} \in \mathcal{S}_i} \ell(h_1, \mathbf{z}) \right) - \left( |\mathcal{S}_i| R_\mu(h_2) - \sum_{\mathbf{z} \in \mathcal{S}_i} \ell(h_2, \mathbf{z}) \right) \right|$$

$$= \left| \left( \sum_{\mathbf{z} \in \mathcal{S}_i} R_\mu(h_1) - \sum_{\mathbf{z} \in \mathcal{S}_i} \ell(h_1, \mathbf{z}) \right) - \left( \sum_{\mathbf{z} \in \mathcal{S}_i} R_\mu(h_2) - \sum_{\mathbf{z} \in \mathcal{S}_i} \ell(h_2, \mathbf{z}) \right) \right|$$

$$\leq \sum_{\mathbf{z} \in \mathcal{S}_i} \mathop{\mathbb{E}}_{\mathbf{z}' \sim \mu} \left[ |\ell(h_1, \mathbf{z}') - \ell(h_2, \mathbf{z}')| + |\ell(h_2, \mathbf{z}) - \ell(h_1, \mathbf{z})| \right]$$

$$\leq \mathop{\mathbb{E}}_{\mathbf{z}' \sim \mu} \sum_{\mathbf{z} \in \mathcal{S}_i} 2L d(h_1, h_2)$$

$$= 2|\mathcal{S}_i| L d(h_1, h_2).$$

We are now able to upper-bound $\mathbb{E}_{h \sim \rho}[R_\mu(h) - \hat{R}_\mathcal{S}(h)]$. Indeed, we have

$$\mathop{\mathbb{E}}_{h \sim \rho} \left[ R_\mu(h) - \hat{R}_\mathcal{S}(h) \right] = \frac{1}{m} \sum_{i=1}^{K} \mathop{\mathbb{E}}_{h \sim \rho} \left[ |\mathcal{S}_i| R_\mu(h) - \sum_{\mathbf{z} \in \mathcal{S}_i} \ell(h, \mathbf{z}) \right]$$

$$\leq \sum_{i=1}^{K} \frac{2|\mathcal{S}_i|L}{m} W(\rho, \pi_i) + \sum_{i=1}^{K} \mathop{\mathbb{E}}_{h \sim \pi_i} \frac{1}{m} \left[ |\mathcal{S}_i| R_\mu(h) - \sum_{\mathbf{z} \in \mathcal{S}_i} \ell(h, \mathbf{z}) \right], \tag{14}$$

where the inequality comes from the Kantorovich-Rubinstein duality theorem. Let $f(\mathcal{S}_i) = \mathbb{E}_{h \sim \pi_i} \frac{1}{m} \left[ |\mathcal{S}_i| R_\mu(h) - \sum_{\mathbf{z} \in \mathcal{S}_i} \ell(h, \mathbf{z}_i) \right]$, the function has the bounded difference inequality, *i.e.*, for two datasets $\mathcal{S}_i$ and $\mathcal{S}_i'$ that differs from one example (the $k$-th example, without loss of generality), we have

$$|f(\mathcal{S}_i) - f(\mathcal{S}_i')| = \left| \mathop{\mathbb{E}}_{h \sim \pi_i} \frac{1}{m} \left[ |\mathcal{S}_i| R_\mu(h) - \sum_{\mathbf{z} \in \mathcal{S}_i} \ell(h, \mathbf{z}) \right] - \mathop{\mathbb{E}}_{h \sim \pi_i} \frac{1}{m} \left[ |\mathcal{S}_i| R_\mu(h) - \sum_{\mathbf{z}' \in \mathcal{S}_i'} \ell(h, \mathbf{z}') \right] \right|$$

$$= \left| \mathop{\mathbb{E}}_{h \sim \pi_i} \left[ \frac{1}{m} |\mathcal{S}_i| R_\mu(h) - \frac{1}{m} \sum_{\mathbf{z} \in \mathcal{S}_i} \ell(h, \mathbf{z}) - \frac{1}{m} |\mathcal{S}_i| R_\mu(h) + \frac{1}{m} \sum_{\mathbf{z}' \in \mathcal{S}_i'} \ell(h, \mathbf{z}') \right] \right|$$

$$= \left| \mathop{\mathbb{E}}_{h \sim \pi_i} \left[ \frac{1}{m} \sum_{\mathbf{z}' \in \mathcal{S}_i'} \ell(h, \mathbf{z}') - \frac{1}{m} \sum_{\mathbf{z} \in \mathcal{S}_i} \ell(h, \mathbf{z}) \right] \right|$$

$$= \left| \mathop{\mathbb{E}}_{h \sim \pi_i} \left[ \frac{1}{m} \ell(h, \mathbf{z}_k') - \frac{1}{m} \ell(h, \mathbf{z}_k) \right] \right|$$

$$\leq \frac{1}{m}.$$

Hence, from Mcdiarmid's inequality, we have with probability at least $1 - \frac{\delta}{K}$ over $\mathcal{S} \sim \mu^m$

$$\mathbb{E}_{h \sim \pi_i} \frac{1}{m} \left[ |\mathcal{S}_i| \mathbf{R}_\mu(h) - \sum_{\mathbf{z} \in \mathcal{S}_i} \ell(h, \mathbf{z}) \right]$$

$$\leq \mathbb{E}_{\mathcal{S} \sim \mu^m} \mathbb{E}_{h \sim \pi_i} \frac{1}{m} \left[ |\mathcal{S}_i| \mathbf{R}_\mu(h) - \sum_{\mathbf{z} \in \mathcal{S}_i} \ell(h, \mathbf{z}) \right] + \sqrt{\frac{|\mathcal{S}_i| \ln \frac{K}{\delta}}{2m^2}}$$

$$= \mathbb{E}_{\mathcal{S}_i^c \sim \mu^{m - |\mathcal{S}_i|}} \mathbb{E}_{\mathcal{S}_i \sim \mu^{|\mathcal{S}_i|}} \mathbb{E}_{h \sim \pi_i} \frac{1}{m} \left[ |\mathcal{S}_i| \mathbf{R}_\mu(h) - \sum_{\mathbf{z} \in \mathcal{S}_i} \ell(h, \mathbf{z}) \right] + \sqrt{\frac{|\mathcal{S}_i| \ln \frac{K}{\delta}}{2m^2}}$$

$$= \mathbb{E}_{\mathcal{S}_i^c \sim \mu^{m - |\mathcal{S}_i|}} \mathbb{E}_{h \sim \pi_i} \frac{1}{m} \left[ |\mathcal{S}_i| \mathbf{R}_\mu(h) - \mathbb{E}_{\mathcal{S}_i \sim \mu^{|\mathcal{S}_i|}} \sum_{\mathbf{z} \in \mathcal{S}_i} \ell(h, \mathbf{z}) \right] + \sqrt{\frac{|\mathcal{S}_i| \ln \frac{K}{\delta}}{2m^2}}$$

$$= \mathbb{E}_{\mathcal{S}_i^c \sim \mu^{m - |\mathcal{S}_i|}} \mathbb{E}_{h \sim \pi_i} \frac{1}{m} \left[ |\mathcal{S}_i| \mathbf{R}_\mu(h) - |\mathcal{S}_i| \mathbf{R}_\mu(h) \right] + \sqrt{\frac{|\mathcal{S}_i| \ln \frac{K}{\delta}}{2m^2}}$$

$$= \sqrt{\frac{|\mathcal{S}_i| \ln \frac{K}{\delta}}{2m^2}}.$$

From the union bound, we have with probability at least $1 - \delta$ over $\mathcal{S} \sim \mu^m$, for any $\rho \in \mathcal{M}(\mathcal{H})$,

$$\mathbb{E}_{h \sim \rho} \left[ \mathbf{R}_\mu(h) - \hat{R}_\mathcal{S}(h) \right] \leq \sum_{i=1}^{K} \frac{2|\mathcal{S}_i|L}{m} \mathbf{W}(\rho, \pi_i) + \sum_{i=1}^{K} \sqrt{\frac{|\mathcal{S}_i| \ln \frac{K}{\delta}}{2m^2}},$$

which is the claimed result. □

We are now able to give a corollary of Theorem 6.

**Corollary 7.** *We assume our loss $\ell$ to be in $[0,1]$ and $L$-Lipschitz. Then, for any $\delta \in (0,1]$, with probability at least $1 - \delta$ over the sample $\mathcal{S}$, the following holds for the hypotheses $h_{i,\mathcal{S}} \in \mathcal{H}$ associated with the Dirac distributions $\pi_{i,\mathcal{S}}$ and for any $h \in \mathcal{H}$:*

$$R_\mu(h) \leq \hat{R}_\mathcal{S}(h) + \sum_{i=1}^{K} \frac{2|\mathcal{S}_i|L}{m} d(h, h_{i,\mathcal{S}}) + \sum_{i=1}^{K} \sqrt{\frac{|\mathcal{S}_i| \ln \frac{K}{\delta}}{2m^2}}.$$

Such a bound was impossible to obtain from the PAC-Bayesian bounds based on a KL divergence. Indeed, the KL divergence is infinite for two distributions with disjoint supports. Hence, the PAC-Bayesian framework based on the Wasserstein distance allows us to provide uniform-convergence bounds from a proof technique different from the ones based on the Rademacher complexity [KP00, BM01, BM02] or the VC-dimension [VC68, VC74]. In Section 4, we provide an algorithm minimising such a bound.

## B.4 Proof of Theorem 3

**Theorem 3.** *We assume our loss $\ell$ to be $L$-Lipschitz. Then, for any $\delta \in (0,1]$, with probability at least $1 - \delta$ over the sample $\mathcal{S}$, the following holds for the distributions $\pi_{i,\mathcal{S}} := \pi_i(\mathcal{S}, .)$ and for any sequence $(\rho_i)_{i=1\cdots m} \in \mathcal{M}(\mathcal{H})^m$:*

$$\sum_{i=1}^{m} \mathbb{E}_{h_i \sim \rho_i} \left[ \mathbb{E}[\ell(h_i, \mathbf{z}_i) \mid \mathcal{F}_{i-1}] - \ell(h_i, \mathbf{z}_i) \right] \leq 2L \sum_{i=1}^{m} \mathbf{W}(\rho_i, \pi_{i,\mathcal{S}})$$

$$+ \frac{\lambda}{2} \sum_{i=1}^{m} \mathbb{E}_{h_i \sim \pi_{i,\mathcal{S}}} \left[ \hat{V}_i(h_i, \mathbf{z}_i) + V_i(h_i) \right] + \frac{\ln(1/\delta)}{\lambda},$$

*where for all $i$, $\hat{V}_i(h_i, \mathbf{z}_i) = (\ell(h_i, \mathbf{z}_i) - \mathbb{E}_{i-1}[\ell(h_i, \mathbf{z}_i)])^2$ is the conditional empirical variance at time $i$ and $V_i(h_i) = \mathbb{E}_{i-1}[\hat{V}(h_i, \mathbf{z}_i)]$ is the true conditional variance.*

*Proof.* First of all, note that for a $L$-Lipschitz loss function $\ell : \mathcal{H} \times \mathcal{Z} \to \mathbb{R}$, we have

$$\left| \left( \mathbb{E}_{i-1}[\ell(h_i, \mathbf{z}_i)] - \ell(h_i, \mathbf{z}_i) \right) - \left( \mathbb{E}_{i-1}[\ell(h_i', \mathbf{z}_i)] - \ell(h_i', \mathbf{z}_i) \right) \right| \leq 2Ld(h_i, h_i'). \tag{15}$$

Indeed, we can deduce Equation (15) from Jensen inequality, the triangle inequality, and by definition that we have

$$\left| \left( \mathbb{E}_{i-1}[\ell(h_i, \mathbf{z}_i)] - \ell(h_i, \mathbf{z}_i) \right) - \left( \mathbb{E}_{i-1}[\ell(h_i', \mathbf{z}_i)] - \ell(h_i', \mathbf{z}_i) \right) \right|$$

$$\leq \mathbb{E}_{i-1} \left[ |\ell(h_i, \mathbf{z}_i') - \ell(h_i', \mathbf{z}_i')| + |\ell(h_i, \mathbf{z}_i) - \ell(h_i', \mathbf{z}_i)| \right]$$

$$\leq \mathbb{E}_{i-1} 2Ld(h_i, h_i') = 2Ld(h_i, h_i').$$

From the Kantorovich-Rubinstein duality theorem [Vil09, Remark 6.5], we have

$$\sum_{i=1}^{m} \mathbb{E}_{h_i \sim \rho_i} \left[ \mathbb{E}_{i-1}[\ell(h_i, \mathbf{z}_i)] - \ell(h_i, \mathbf{z}_i) \right] \leq 2L \sum_{i=1}^{m} W_1(\rho_i, \pi_{i,\mathcal{S}}) + \sum_{i=1}^{m} \mathbb{E}_{h \sim \pi_{i,\mathcal{S}}} \left[ R_\mu(h_i) - \ell(h_i, \mathbf{z}_i) \right].$$

Now, we define $X_i(h_i, \mathbf{z}_i) := \mathbb{E}_{i-1}[\ell(h_i, \mathbf{z}_i)] - \ell(h_i, \mathbf{z}_i)$. We also recall that for any $i$, we have $\hat{V}_i(h_i, \mathbf{z}_i) = (\ell(h_i, \mathbf{z}_i) - \mathbb{E}_{i-1}[\ell(h_i, \mathbf{z}_i)])^2$ and $V_i(h_i) = \mathbb{E}_{i-1}[\hat{V}(h_i, \mathbf{z}_i)]$. To apply the supermartingales techniques of [HG23a], we define the following function:

$$f_m(S, h_1, ..., h_m) := \sum_{i=1}^{m} \lambda X_i(h_i, \mathbf{z}_i) - \frac{\lambda^2}{2} \sum_{i=1}^{m} (\hat{V}_i(h_i, \mathbf{z}_i) + V_i(h_i)).$$

Now, Lemma 3.2 of [HG23a] state that the sequence $(SM_m)_{m \geq 1}$ defined for any $m$ as:

$$SM_m := \mathbb{E}_{(h_1, \cdots, h_m) \sim \pi_{1,\mathcal{S}} \otimes \cdots \otimes \pi_{m,\mathcal{S}}} \left[ \exp \left( f_m(\mathcal{S}, h_1, ..., h_m) \right) \right],$$

is a supermartingale. We exploit this fact as follows:

$$\sum_{i=1}^{m} \mathbb{E}_{h \sim \rho_{i-1}} \left[ \mathbb{E}_{i-1}[\ell(h_i, \mathbf{z}_i)] - \ell(h_i, \mathbf{z}_i) \right] = \mathbb{E}_{(h_1, \cdots, h_m) \sim \pi_{1,\mathcal{S}} \otimes \cdots \otimes \pi_{m,\mathcal{S}}} \left[ \sum_{i=1}^{m} X_i(h_i, \mathbf{z}_i) \right]$$

$$= \frac{1}{\lambda} \mathbb{E}_{(h_1, \cdots, h_m) \sim \pi_{1,\mathcal{S}} \otimes \cdots \otimes \pi_{m,\mathcal{S}}} [f_m(\mathcal{S}, h_1, \cdots, h_m)]$$

$$+ \frac{\lambda}{2} \sum_{i=1}^{m} \mathbb{E}_{h_i \sim \pi_{i,\mathcal{S}}} \left[ \hat{V}_i(h_i, \mathbf{z}_i) + V_i(h_i) \right]$$

$$\leq \frac{\ln(SM_m)}{\lambda} + \frac{\lambda}{2} \sum_{i=1}^{m} \mathbb{E}_{h_i \sim \pi_{i,\mathcal{S}}} \left[ \hat{V}_i(h_i, \mathbf{z}_i) + V_i(h_i) \right]$$

The last line holds thanks to Jensen's inequality. Now using Ville's inequality ensures us that:

$$\mathbb{P}_{\mathcal{S}} \left( \forall m, SM_m \leq \frac{1}{\delta} \right) \geq \frac{1}{\delta}.$$

Thus, with probability $1 - \delta$, for any $m$ we have $\ln(SM_m) \leq \ln\left(\frac{1}{\delta}\right)$. This concludes the proof. $\square$

### B.5 Proof of Theorem 4

**Theorem 4.** *We assume our loss $\ell$ to be non-negative and $L$-Lipschitz. We also assume that, for any $i, \mathcal{S}, \mathbb{E}_{h \sim \pi_i(.,\mathcal{S})} \left[ \mathbb{E}_{i-1}[\ell(h, \mathbf{z}_i)^2] \right] \leq 1$ (bounded conditional order 2 moments for priors). Then, for any $\delta \in (0, 1]$, with probability at least $1 - \delta$ over the sample $\mathcal{S}$, any online predictive sequence (used as priors) $(\pi_i)_{i \geq 1}$, we have with probability at least $1 - \delta$ over the sample $S \sim \mu$, the following, holding for the data-dependent measures $\pi_{i,\mathcal{S}} := \pi_i(S, .)$ and any posterior sequence $(\rho_i)_{i \geq 1}$:*

$$\frac{1}{m} \sum_{i=1}^{m} \mathbb{E}_{h_i \sim \rho_i} \left[ \mathbb{E}[\ell(h_i, \mathbf{z}_i) \mid \mathcal{F}_{i-1}] - \ell(h_i, \mathbf{z}_i) \right] \leq \frac{2L}{m} \sum_{i=1}^{m} W(\rho_i, \pi_{i,\mathcal{S}}) + \sqrt{\frac{2 \ln\left(\frac{1}{\delta}\right)}{m}}.$$

*Proof.* The proof starts similarly to the one of Theorem 3. Indeed, note that for a $L$-Lipschitz loss function $\ell : \mathcal{H} \times \mathcal{Z} \to \mathbb{R}$, we have

$$\left| \left( \underset{i-1}{\mathbb{E}} [\ell(h_i, \mathbf{z}_i)] - \ell(h_i, \mathbf{z}_i) \right) - \left( \underset{i-1}{\mathbb{E}} [\ell(h'_i, \mathbf{z}_i)] - \ell(h'_i, \mathbf{z}_i) \right) \right| \leq 2Ld(h_i, h'_i). \tag{16}$$

Indeed, we can deduce Equation (16) from Jensen inequality, the triangle inequality, and by definition that we have

$$\left| \left( \underset{i-1}{\mathbb{E}} [\ell(h_i, \mathbf{z}_i)] - \ell(h_i, \mathbf{z}_i) \right) - \left( \underset{i-1}{\mathbb{E}} [\ell(h'_i, \mathbf{z}_i)] - \ell(h'_i, \mathbf{z}_i) \right) \right|$$
$$\leq \underset{i-1}{\mathbb{E}} \left[ |\ell(h_i, \mathbf{z}'_i) - \ell(h'_i, \mathbf{z}'_i)| + |\ell(h_i, \mathbf{z}_i) - \ell(h'_i, \mathbf{z}_i)| \right]$$
$$\leq \underset{i-1}{\mathbb{E}} \, 2Ld(h_i, h'_i) = 2Ld(h_i, h'_i).$$

From the Kantorovich-Rubinstein duality theorem [Vil09, Remark 6.5], we have

$$\sum_{i=1}^{m} \underset{h_i \sim \rho_i}{\mathbb{E}} \left[ \underset{i-1}{\mathbb{E}} [\ell(h_i, \mathbf{z}_i)] - \ell(h_i, \mathbf{z}_i) \right] \leq 2L \sum_{i=1}^{m} W_1(\rho_i, \pi_{i,\mathcal{S}}) + \sum_{i=1}^{m} \underset{h \sim \pi_{i,\mathcal{S}}}{\mathbb{E}} \left[ \mathbf{R}_\mu(h_i) - \ell(h_i, \mathbf{z}_i) \right].$$

Now, we define $X_i(h_i, \mathbf{z}_i) := \mathbb{E}_{i-1}[\ell(h_i, \mathbf{z}_i)] - \ell(h_i, \mathbf{z}_i)$. To apply the supermartingales techniques of [CWR23], we define the following function:

$$f_m(S, h_1, ..., h_m) := \sum_{i=1}^{m} \lambda X_i(h_i, \mathbf{z}_i) - \frac{\lambda^2}{2} \sum_{i=1}^{m} \underset{i-1}{\mathbb{E}} [\ell(h_i, \mathbf{z}_i)^2].$$

Now, because our loss is nonnegative, [CWR23, Lemma A.2 and Lemma B.1] state that the sequence $(SM_m)_{m \geq 1}$ defined for any $m$ as:

$$SM_m := \underset{(h_1, \cdots, h_m) \sim \pi_{1,\mathcal{S}} \otimes \cdots \otimes \pi_{m,\mathcal{S}}}{\mathbb{E}} \left[ \exp \left( f_m(\mathcal{S}, h_1, ..., h_m) \right) \right],$$

is a supermartingale. We exploit this fact as follows:

$$\sum_{i=1}^{m} \underset{h \sim \rho_{i-1}}{\mathbb{E}} \left[ \underset{i-1}{\mathbb{E}} [\ell(h_i, \mathbf{z}_i)] - \ell(h_i, \mathbf{z}_i) \right] = \underset{(h_1, \cdots, h_m) \sim \pi_{1,\mathcal{S}} \otimes \cdots \otimes \pi_{m,\mathcal{S}}}{\mathbb{E}} \left[ \sum_{i=1}^{m} X_i(h_i, \mathbf{z}_i) \right]$$
$$= \frac{1}{\lambda} \underset{(h_1, \cdots, h_m) \sim \pi_{1,\mathcal{S}} \otimes \cdots \otimes \pi_{m,\mathcal{S}}}{\mathbb{E}} [f_m(\mathcal{S}, h_1, \cdots, h_m)]$$
$$+ \frac{\lambda}{2} \sum_{i=1}^{m} \underset{h_i \sim \pi_{i,\mathcal{S}}}{\mathbb{E}} \left[ \underset{i-1}{\mathbb{E}} [\ell(h_i, \mathbf{z}_i)^2] \right]$$
$$\leq \frac{\ln (SM_m)}{\lambda} + \frac{\lambda}{2} \sum_{i=1}^{m} \underset{h_i \sim \pi_{i,\mathcal{S}}}{\mathbb{E}} \left[ \underset{i-1}{\mathbb{E}} [\ell(h_i, \mathbf{z}_i)^2] \right]$$

The last line holds thanks to Jensen's inequality. Now using Ville's inequality ensures us that:

$$\underset{\mathcal{S}}{\mathbb{P}} \left( \forall m, SM_m \leq \frac{1}{\delta} \right) \geq \frac{1}{\delta}$$

Thus, with probability $1-\delta$, for any $m$ we have $\ln(SM_m) \leq \ln \frac{1}{\delta}$. We conclude the proof by exploiting the boundedness assumption on conditional order 2 moments and optimising the bound in $\lambda$. $\square$

## C    Supplementary insights on experiments

In this section, Appendix C.1 presents the learning algorithm for the *i.i.d.* setting. We also introduce the online algorithm in Appendix C.2. We prove the Lipschitz constant of the loss for the linear models in Appendix C.3. Finally, we provide more experiments in Appendix C.5.

## C.1 Batch algorithm for the *i.i.d.* setting

The pseudocode of our batch algorithm is presented in Algorithm 1.

---

**Algorithm 1** (Mini-)Batch Learning Algorithm with Wasserstein distances

---

1: **procedure** PRIORS LEARNING
2:     $h_1, \ldots, h_K \leftarrow$ initialize the hypotheses
3:     **for** $t \leftarrow 1, \ldots, T$ **do**
4:         **for each** mini-batch $\mathcal{U} \subseteq \mathcal{S}$ **do**
5:             **for** $i \leftarrow 1, \ldots, K$ **do**
6:                 $\mathcal{U}_i \leftarrow \mathcal{U} \setminus \mathcal{S}_i$
7:                 $h_i \leftarrow$ perform a gradient descent step with $\nabla \mathrm{R}_{\mathcal{U}_i}(h_i)$
8:     **return** hypotheses $h_1, \ldots, h_K$

9:
10: **procedure** POSTERIOR LEARNING
11:     $h \leftarrow$ initialize the hypothesis
12:     **for** $t \leftarrow 1, \ldots, T'$ **do**
13:         **for each** mini-batch $\mathcal{U} \subseteq \mathcal{S}$ **do**
14:             $h \leftarrow$ perform a gradient descent step with $\nabla[\mathrm{R}_{\mathcal{U}}(h) + \varepsilon \sum_{i=1}^{K} \frac{|\mathcal{S}_i|}{m} d(h, h_i)]$
15:     **return** hypothesis $h$

---

PRIORS LEARNING minimises the empirical risk through mini-batches $\mathcal{U} \subseteq \mathcal{S}$ for $T$ epochs. More precisely, for each epoch, we *(a)* sample a mini-batch $\mathcal{U}$ (line 4) by excluding the set $\mathcal{S}_i$ from $\mathcal{U}$ for each $h_i \in \mathcal{H}$ (line 5-6), then *(b)* the hypotheses $h_1, \ldots, h_K \in \mathcal{H}$ are updated (line 7). In POSTERIOR LEARNING, we perform a gradient descent step (line 14) on the objective function associated with Equation (5) for $T'$ epochs in a mini-batch fashion.

## C.2 Learning algorithm for the online setting

Algorithm 2 presents the pseudocode of our online algorithm.

---

**Algorithm 2** Online Learning Algorithm with Wasserstein distances

---

1: Initialize the hypothesis $h_0 \in \mathcal{H}$
2: **for** $i \leftarrow 1, \ldots, m$ **do**
3:     **for** $t \leftarrow 1, \ldots, T$ **do**
4:         $h_i \leftarrow$ perform a gradient step with $\nabla[\ell(h_i, \mathbf{z}_i) + \hat{B}(d(h_i, h_{i-1}) - 1)]$ (Eq. (7) with $\hat{B}$)
5: **return** hypotheses $h_1, \ldots, h_m$

---

For each time step $i$, we perform $T$ gradient descent steps on the objective associated with Equation (6) (line 4). Note that we can retrieve OGD from Algorithm 2 by *(a)* setting $T = 1$ and *(b)* removing the regularisation term $\hat{B}(d(h_i, h_{i-1}) - 1)$.

## C.3 Lipschitzness for the linear model

Recall that we use, in our experiments, the multi-margin loss function from the Pytorch module defined for any linear model with weights $W \in \mathbb{R}^{|\mathcal{Y}| \times d}$ and biases $b \in \mathbb{R}^{|\mathcal{Y}|}$, any data point $\mathbf{z} \in \mathcal{X} \times \mathcal{Y}$

$$\ell(W, b, \mathbf{z}) = \frac{1}{|\mathcal{Y}| - 1} \sum_{y' \neq y} \max\left(0, f(W, b, \mathbf{z}, y')\right),$$

where $f(W, b, \mathbf{z}, y') = 1 + \langle W[y'] - W[y], \mathbf{x} \rangle + b[y'] - b[y]$, and $W[y] \in \mathbb{R}^d$ and $b[y] \in \mathbb{R}$ are respectively the vector and the scalar for the $y$-th output.

To apply our theorems, we must ensure that our loss function is Lipschitz with respect to the linear model, hence the following lemma.

**Lemma 8.** *For any $\mathbf{z} = (\mathbf{x}, y) \in \mathcal{X} \times \mathcal{Y}$ with the norm of $\mathbf{x}$ bounded by 1, the function $W, b \mapsto \ell(W, b, \mathbf{z})$ is 2-Lipschitz.*

*Proof.* Let $(W, b), (W', b')$ both in $\mathbb{R}^{|\mathcal{Y}| \times d} \times \mathbb{R}^{|\mathcal{Y}|}$, we have

$$|\ell(W, b, \mathbf{z}) - \ell(W', b', \mathbf{z})| \leq \frac{1}{|\mathcal{Y}| - 1} \sum_{y' \neq y} |\max(0, f(W, b, \mathbf{z}, y')) - \max(0, f(W', b', \mathbf{z}, y'))|.$$

Note that because $\alpha \mapsto \max(0, \alpha)$ is 1-Lipschitz, we have:

$$|\ell(W, b, \mathbf{z}) - \ell(W', b', \mathbf{z})| \leq \frac{1}{|\mathcal{Y}| - 1} \sum_{y' \neq y} |f(W, b, \mathbf{z}, y') - f(W', b', \mathbf{z}, y')|.$$

Finally, notice that:

$$\frac{1}{|\mathcal{Y}| - 1} \sum_{y' \neq y} |f(W, b, \mathbf{z}, y') - f(W', b', \mathbf{z}, y')| \leq \frac{1}{|\mathcal{Y}| - 1} \sum_{y' \neq y} |\langle (W - W')[y'] - (W - W')[y], \mathbf{x} \rangle|$$

$$+ \frac{1}{|\mathcal{Y}| - 1} \sum_{y' \neq y} |(b - b')[y'] - (b - b')[y]|$$

$$\leq \frac{1}{|\mathcal{Y}| - 1} \sum_{y' \neq y} \|(W - W')[y'] - (W - W')[y]\| \, \|\mathbf{x}\|$$

$$+ \frac{1}{|\mathcal{Y}| - 1} \sum_{y' \neq y} |(b - b')[y'] - (b - b')[y]|.$$

Because we consider the Euclidean norm, we have for any $y' \in \mathcal{Y}$:

$$\|(W - W')[y'] - (W - W')[y]\| = \sqrt{\|(W - W')[y'] - (W - W')[y]\|^2}$$

$$\leq \sqrt{2 \left( \|(W - W')[y']\|^2 + \|(W - W')[y]\|^2 \right)}$$

$$\leq \sqrt{2}\|W - W'\|.$$

The second line holding because for any scalars $a, b$, we have $(a - b)^2 \leq 2(a^2 + b^2)$ and the last line holding because $\|W - W'\|^2 = \sum_{y \in \mathcal{Y}} \|(W - W')[y]\|^2$. A similar argument gives

$$\frac{1}{|\mathcal{Y}| - 1} \sum_{y' \neq y} |(b - b')[y'] - (b - b')[y]| \leq \sqrt{2}\|b - b'\|.$$

Then, using that $\|x\| \leq 1$ and summing on all $y'$ gives:

$$|\ell(W, b, \mathbf{z}) - \ell(W', b', \mathbf{z})| \leq \sqrt{2} \left( \|W - W'\| + \|b - b'\| \right).$$

Finally, notice that $(\|W - W'\| + \|b - b'\|)^2 \leq 2(\|W - W'\|^2 + \|b - b'\|^2) = 2\|(W, b) - (W', b')\|^2$.

Thus $\|W - W'\| + \|b - b'\| \leq \sqrt{2}\|(W, b) - (W', b')\|$. This concludes the proof. $\square$

### C.4  Lipschitzness for neural networks

Recall that we use, in our experiments, the multi-margin loss function from the Pytorch module defined we consider the loss $\ell(h, (\mathbf{x}, y)) = \frac{1}{|\mathcal{Y}|} \sum_{y' \neq y} \max(0, 1 - \eta(h[y] - h[y']))$, which is $\eta$-Lipschitz *w.r.t.* the outputs $h[1], \dots, h[|\mathcal{Y}|]$. For neural networks, $h$ is the output of the neural network with input $\mathbf{x}$. Note that this loss is $\eta$-lipschitz with respect to the outputs. To apply our theorems, we must ensure that our loss function is Lipschitz with respect to the weights of the neural networks, hence the following lemma with associated background.

We define a FCN recursively as follows: for a vector $\mathbf{W}_1 = \text{vec}(\{W_1, b\})$, (*i.e.*, the vectorisation of a weight matrix $W_1$ and a bias $b$) and an input datum $\mathbf{x}$, $\text{FCN}_1(\mathbf{W}_1, \mathbf{x}) = \sigma_1(W_1\mathbf{x} + b_1)$, where $\sigma_1$ is the activation function. Also, for any $i \geq 2$ we define for a vector $\mathbf{W}_i = (W_i, b_i, \mathbf{W}_{i-1})$ (defined recursively as well), $\text{FCN}_i(\mathbf{W}_i, \mathbf{x}) = \sigma_i(W_i\text{FCN}_{i-1}(\mathbf{W}_{i-1}, \mathbf{x}) + b_i)$. Then, setting $\mathbf{z} = (\mathbf{x}, y)$ a datum and $h_i(\mathbf{x}) := \text{FCN}_i(\mathbf{W}_i, \mathbf{x})$ we can rewrite our loss as a function of $(\mathbf{W}_i, \mathbf{z})$.

**Lemma 9.** *Assume that all the weight matrices of $\mathbf{W}_i$ are bounded and that the activation functions are Lipschitz continuous with constant bounded by $K_\sigma$. Then for any datum $\mathbf{z} = (\mathbf{x}, y)$, any $i$, $\mathbf{W}_i \to \ell(\mathbf{W}_i, \mathbf{z})$ is Lipschitz continuous.*

*Proof.* We consider the Frobenius norm on matrices as $\mathbf{W}_2$ is a vector as we consider the L2-norm on the vector. We prove the result for $i = 2$, assuming it is true for $i = 1$. We then explain how this proof generalises the case $i = 1$ and works recursively. Let $\mathbf{z}, \mathbf{W}_2, \mathbf{W}_2'$, for clarity we write $\text{FCN}_2(\mathbf{x}) := \text{FCN}(\mathbf{W}_2, \mathbf{x})$ and $\text{FCN}_2'(\mathbf{x}) := \text{FCN}(\mathbf{W}_2', \mathbf{x})$. As $\ell$ is Lipschitz on the outputs $\text{FCN}_2(\mathbf{x}), \text{FCN}_2'(\mathbf{x})$. We have

$$
\begin{aligned}
|\ell(\mathbf{W}_2, \mathbf{z}) - \ell(\mathbf{W}_2', \mathbf{z})| &\leq \eta \left\|\text{FCN}_2(\mathbf{x}) - \text{FCN}_2'(\mathbf{x})\right\| \\
&\leq \eta \left\|\sigma_2\left(W_2\text{FCN}_1(\mathbf{x}) + b_2\right) - \sigma_2\left(W_2'\text{FCN}_1'(\mathbf{x}) + b_2'\right)\right\| \\
&\leq \eta K_\sigma \|W_2\text{FCN}_1(\mathbf{x}) + b_2 - W_2'\text{FCN}_1'(\mathbf{x}) - b_2'\| \\
&\leq \eta K_\sigma \left(||(W_2 - W_2')\text{FCN}_1(\mathbf{x})|| + ||W_2'(\text{FCN}_1(\mathbf{x}) - \text{FCN}_1'(\mathbf{x}))|| + \|b_2 - b_2'\|\right).
\end{aligned}
$$

Then, we have $||(W_2 - W_2')\text{FCN}_1(\mathbf{x})|| \leq ||(W_2 - W_2')||_F||\text{FCN}_1(\mathbf{x})|| \leq K_\mathbf{x}||(W_2 - W_2')||_F$. The second inequality holding as $\text{FCN}_1(\mathbf{x})$ is a continuous function of the weights. Indeed, as on a compact space, a continuous function reaches its maximum, then its norm is bounded by a certain $K_\mathbf{x}$. Also, as the weights are bounded, any weight matrix has its norm bounded by a certain $K_W$ thus $\|W_2'(\text{FCN}_1(\mathbf{x}) - \text{FCN}_1'(\mathbf{x})\| \leq \|W_2'\|_F\|(\text{FCN}_1(\mathbf{x}) - \text{FCN}_1'(\mathbf{x})\| \leq K_W\|\text{FCN}_1(\mathbf{x}) - \text{FCN}_1'(\mathbf{x})\|$. Finally, taking $K_{\text{temp}} = \eta K_\sigma \max(K_\mathbf{x}, K_W, 1)$ gives:

$$
|\ell(\mathbf{W}_2, \mathbf{z}) - \ell(\mathbf{W}_2', \mathbf{z})| \leq K_{\text{temp}}\left(\|(W_2 - W_2')\|_F + \|b_2 - b_2'\| + \|\text{FCN}_1(\mathbf{x}) - \text{FCN}_1'(\mathbf{x})\|\right).
$$

Exploiting the recursive assumption that $\text{FCN}_1$ is Lipschitz with respect to its weights $\mathbf{W}_1$ gives $\|\text{FCN}_1(\mathbf{x}) - \text{FCN}_1'(\mathbf{x})\| \leq K_1\|\mathbf{W}_1 - \mathbf{W}_1'\|$.

If we denote by $(W_2, b_2)$ the vector of all concatenated weights, notice that $\|(W_2 - W_2')\|_F + \|b_2 - b_2'\| = \sqrt{(\|(W_2 - W_2')\|_F + \|b_2 - b_2'\|)^2} \leq \sqrt{2(\|(W_2 - W_2')\|_F^2 + \|b_2 - b_2'\|^2)} = \sqrt{2}\|(W_2, b_2) - (W_2', b_2')\|$ (we used that for any real numbers $a, b, (a+b)^2 \leq 2(a^2 + b^2)$). We then have:

$$
\begin{aligned}
|\ell(\mathbf{W}_2, \mathbf{z}) - \ell(\mathbf{W}_2', \mathbf{z})| &\leq K_{\text{temp}}\max(\sqrt{2}, K_1)\left(\|(W_2, b_2) - (W_2', b_2')\| + ||\mathbf{W}_1 - \mathbf{W}_1'||\right) \\
&\leq \sqrt{2}K_{\text{temp}}\max(\sqrt{2}, K_1)||\mathbf{W}_2 - \mathbf{W}_2'||.
\end{aligned}
$$

The last line holds by reusing the same calculation trick. This concludes the proof for $i = 2$. Then for $i = 1$ the same proof holds by replacing $W_2, b_2, \text{FCN}_2$ by $W_1, b_1, \text{FCN}_1$ and replacing $\text{FCN}_1(\mathbf{x}), \text{FCN}_1'(\mathbf{x})$ by $\mathbf{x}$ (we then do not need to assume a recursive Lipschitz behaviour). Therefore the result holds for $i = 1$.

We then properly apply a recursive argument by assuming the result at rank $i - 1$ reusing the same proof at any rank $i$ by replacing $W_2, b_2, \text{FCN}_2$ by $W_i, b_i, \text{FCN}_i$ and $\text{FCN}_1(\mathbf{x}), \text{FCN}_1'(\mathbf{x})$ by $\text{FCN}_{i-1}(\mathbf{x}), \text{FCN}_{i-1}'(\mathbf{x})$. This concludes the proof. $\square$

## C.5 Experiments with varying number of priors

The experiments of Section 4 rely on data-dependent priors constructed through the procedure PRIORS LEARNING. We fixed a number of priors $K$ equal to $0.2\sqrt{m}$. This number is an empirical tradeoff between the informativeness of our priors and time-efficient computation. However, there is no theoretical intuition for the value of this parameter (the discussion of Section 3.1 considered $K = \sqrt{m}$ as a potential tradeoff; see Appendix A). Thus, we gather below the performance of our learning procedures for $K = \alpha\sqrt{m}$, where $\alpha \in \{0, 0.4, 0.6, 0.8, 1\}$ (the case $\alpha = 0$ being a convention to

denote $K = 1$). The experiments are gathered below, and all remaining hyperparameters (except $K$) are identical to those described in Section 4.

**Analysis of our results.** First, when considering neural networks, note that for any dataset except SEGMENTATION, LETTER, the performances of our methods are similar or better when considering data-dependent priors (*i.e.*, when $\alpha > 0$). A similar remark holds for the linear models for all datasets except for SATIMAGE, SEGMENTATION, and TICTACTOE. This illustrates the relevance of data-dependent priors. We also remark that there is no value of $\alpha$, which provides a better performance on all datasets. For instance, considering neural networks, note that $\alpha = 1$ gives the better performance (*i.e.*, the smallest $\mathfrak{R}_\mu(h)$) for Algorithm 1 ($1/\sqrt{m}$) for the SATIMAGE dataset while, for the same algorithm, the better performance on the SEGMENTATION dataset is attained for $\alpha = 0.8$. Sometimes, the number $K$ does not have a clear influence: on MNIST with NNs, for Algorithm 1 ($1/\sqrt{m}$), our performances are similar, whatever the value of $K$, but still significantly better than ERM. In any case, note that for every dataset, there exists a value of $K$ and such that our algorithm attains either similar or significantly better performances than ERM on every dataset, which shows the relevance of our learning algorithm to ensure a good generalisation ability. Moreover, there is no obvious choice for the parameters $\varepsilon$. For instance, in Tables 2 and 3, for the SEGMENTATION dataset, the parameters $K = 1, \varepsilon = \frac{1}{m}$ are optimal (in terms of test risks) for both models. As $K = 1$ means that our single prior is data-free, this shows that the intrinsic structure of SEGMENTATION makes it less sensitive to both the information contained in the prior ($K = 1$ meaning data-free prior) and the place of the prior itself ($\varepsilon = 1/m$ meaning that we give less weight to the regularisation within our optimisation procedure). On the contrary, in Table Table 1, the YEAST dataset performs significantly better when $\varepsilon = 1/\sqrt{m}(K = 0.2\sqrt{m})$, exhibiting a positive impact of our data-dependent priors.

### C.6 Experiments on classical regularisation methods

We perform additional experiments to see the performance of the weight decay, *i.e.*, the L2 regularisation on the weights; the results are presented in Table 4. Moreover, notice that the 'distance to initialisation' $\|\mathbf{w} - \mathbf{w}_0\|$ (where $\mathbf{w}_0$ is the weights initialized randomly) is a particular case of Algorithm 1 when $K = 1$ (*i.e.*, we treat the data as a single batch, and the prior is the data-free initialisation); the results are in Tables 2 and 3.

**Analysis of our results.** This experiment on the weight decay demonstrates that on a few datasets (namely SENSORLESS and YEAST), when our predictors are neural nets, the weight decay regularisation fails to learn while ours succeeds, as shown in Table 1. In general, this table shows that, on most of the datasets, considering data-dependent priors leads to sharper results. This shows the efficiency of our method compared to the 'distance to initialisation' regularisation.

Table 2: Performance of Algorithm 1 compared to ERM on different datasets for neural network models. We consider $\varepsilon = 1/m$ and $\varepsilon = 1/\sqrt{m}$, with $K = \alpha\sqrt{m}$ and $\alpha \in \{0, 0.4, 0.6, 0.8, 1\}$. We plot the empirical risk $\mathfrak{R}_S(h)$ with its associated test risk $\mathfrak{R}_\mu(h)$.

(a) $K = 1$

| Dataset | Alg. 1 ($\frac{1}{m}$) | | Alg. 1 ($\frac{1}{\sqrt{m}}$) | |
|---|---|---|---|---|
| | $\mathfrak{R}_S(h)$ | $\mathfrak{R}_\mu(h)$ | $\mathfrak{R}_S(h)$ | $\mathfrak{R}_\mu(h)$ |
| ADULT | 0.207 | 0.207 | 0.248 | 0.248 |
| FASHIONMNIST | 0.160 | 0.164 | 0.158 | 0.164 |
| LETTER | 0.258 | 0.269 | 0.268 | 0.280 |
| MNIST | 0.116 | 0.123 | 0.085 | 0.096 |
| MUSHROOMS | 0.000 | 0.000 | 0.000 | 0.001 |
| NURSERY | 0.705 | 0.720 | 0.720 | 0.736 |
| PENDIGITS | 0.704 | 0.724 | 0.021 | 0.037 |
| PHISHING | 0.048 | 0.052 | 0.038 | 0.055 |
| SATIMAGE | 0.148 | 0.208 | 0.147 | 0.207 |
| SEGMENTATION | 0.141 | 0.176 | 0.248 | 0.385 |
| SENSORLESS | 0.907 | 0.911 | 0.907 | 0.911 |
| TICTACTOE | 0.000 | 0.042 | 0.000 | 0.033 |
| YEAST | 0.695 | 0.712 | 0.677 | 0.658 |

(b) $K = 0.4\sqrt{m}$

| Dataset | Alg. 1 ($\frac{1}{m}$) | | Alg. 1 ($\frac{1}{\sqrt{m}}$) | |
|---|---|---|---|---|
| | $\mathfrak{R}_S(h)$ | $\mathfrak{R}_\mu(h)$ | $\mathfrak{R}_S(h)$ | $\mathfrak{R}_\mu(h)$ |
| ADULT | 0.167 | 0.166 | 0.164 | 0.164 |
| FASHIONMNIST | 0.160 | 0.164 | 0.156 | 0.160 |
| LETTER | 0.263 | 0.275 | 0.252 | 0.263 |
| MNIST | 0.112 | 0.120 | 0.085 | 0.096 |
| MUSHROOMS | 0.000 | 0.000 | 0.000 | 0.000 |
| NURSERY | 0.705 | 0.720 | 0.706 | 0.719 |
| PENDIGITS | 0.011 | 0.025 | 0.010 | 0.022 |
| PHISHING | 0.043 | 0.053 | 0.041 | 0.052 |
| SATIMAGE | 0.147 | 0.178 | 0.145 | 0.174 |
| SEGMENTATION | 0.345 | 0.408 | 0.225 | 0.416 |
| SENSORLESS | 0.075 | 0.078 | 0.074 | 0.077 |
| TICTACTOE | 0.000 | 0.031 | 0.000 | 0.019 |
| YEAST | 0.450 | 0.480 | 0.695 | 0.712 |

(c) $K = 0.6\sqrt{m}$

| Dataset | Alg. 1 ($\frac{1}{m}$) | | Alg. 1 ($\frac{1}{\sqrt{m}}$) | |
|---|---|---|---|---|
| | $\mathfrak{R}_S(h)$ | $\mathfrak{R}_\mu(h)$ | $\mathfrak{R}_S(h)$ | $\mathfrak{R}_\mu(h)$ |
| ADULT | 0.165 | 0.163 | 0.165 | 0.164 |
| FASHIONMNIST | 0.158 | 0.164 | 0.156 | 0.160 |
| LETTER | 0.259 | 0.275 | 0.260 | 0.267 |
| MNIST | 0.112 | 0.121 | 0.084 | 0.094 |
| MUSHROOMS | 0.000 | 0.000 | 0.000 | 0.000 |
| NURSERY | 0.706 | 0.719 | 0.706 | 0.719 |
| PENDIGITS | 0.008 | 0.023 | 0.009 | 0.022 |
| PHISHING | 0.043 | 0.055 | 0.040 | 0.050 |
| SATIMAGE | 0.138 | 0.184 | 0.141 | 0.174 |
| SEGMENTATION | 0.577 | 0.845 | 0.145 | 0.309 |
| SENSORLESS | 0.073 | 0.076 | 0.073 | 0.076 |
| TICTACTOE | 0.000 | 0.023 | 0.000 | 0.013 |
| YEAST | 0.461 | 0.449 | 0.410 | 0.426 |

(d) $K = 0.8\sqrt{m}$

| Dataset | Alg. 1 ($\frac{1}{m}$) | | Alg. 1 ($\frac{1}{\sqrt{m}}$) | |
|---|---|---|---|---|
| | $\mathfrak{R}_S(h)$ | $\mathfrak{R}_\mu(h)$ | $\mathfrak{R}_S(h)$ | $\mathfrak{R}_\mu(h)$ |
| ADULT | 0.165 | 0.164 | 0.164 | 0.164 |
| FASHIONMNIST | 0.158 | 0.163 | 0.156 | 0.161 |
| LETTER | 0.260 | 0.274 | 0.260 | 0.267 |
| MNIST | 0.113 | 0.121 | 0.083 | 0.093 |
| MUSHROOMS | 0.000 | 0.000 | 0.000 | 0.000 |
| NURSERY | 0.706 | 0.719 | 0.704 | 0.721 |
| PENDIGITS | 0.011 | 0.026 | 0.008 | 0.020 |
| PHISHING | 0.042 | 0.054 | 0.048 | 0.063 |
| SATIMAGE | 0.136 | 0.174 | 0.128 | 0.183 |
| SEGMENTATION | 0.140 | 0.463 | 0.121 | 0.249 |
| SENSORLESS | 0.075 | 0.079 | 0.074 | 0.077 |
| TICTACTOE | 0.392 | 0.301 | 0.000 | 0.008 |
| YEAST | 0.394 | 0.422 | 0.686 | 0.671 |

(e) $K = \sqrt{m}$

| Dataset | Alg. 1 ($\frac{1}{m}$) | | Alg. 1 ($\frac{1}{\sqrt{m}}$) | |
|---|---|---|---|---|
| | $\mathfrak{R}_S(h)$ | $\mathfrak{R}_\mu(h)$ | $\mathfrak{R}_S(h)$ | $\mathfrak{R}_\mu(h)$ |
| ADULT | 0.167 | 0.166 | 0.167 | 0.167 |
| FASHIONMNIST | 0.159 | 0.164 | 0.157 | 0.162 |
| LETTER | 0.253 | 0.269 | 0.255 | 0.267 |
| MNIST | 0.110 | 0.119 | 0.084 | 0.094 |
| MUSHROOMS | 0.000 | 0.000 | 0.000 | 0.000 |
| NURSERY | 0.706 | 0.719 | 0.703 | 0.722 |
| PENDIGITS | 0.012 | 0.026 | 0.012 | 0.024 |
| PHISHING | 0.042 | 0.050 | 0.039 | 0.053 |
| SATIMAGE | 0.137 | 0.176 | 0.125 | 0.169 |
| SEGMENTATION | 0.162 | 0.398 | 0.153 | 0.387 |
| SENSORLESS | 0.075 | 0.077 | 0.073 | 0.076 |
| TICTACTOE | 0.000 | 0.035 | 0.000 | 0.008 |
| YEAST | 0.415 | 0.435 | 0.418 | 0.450 |

(f) ERM

| Dataset | $\mathfrak{R}_S(h)$ | $\mathfrak{R}_\mu(h)$ |
|---|---|---|
| ADULT | 0.165 | 0.163 |
| FASHIONMNIST | 0.163 | 0.167 |
| LETTER | 0.258 | 0.270 |
| MNIST | 0.119 | 0.127 |
| MUSHROOMS | 0.000 | 0.000 |
| NURSERY | 0.706 | 0.719 |
| PENDIGITS | 0.009 | 0.022 |
| PHISHING | 0.046 | 0.055 |
| SATIMAGE | 0.141 | 0.189 |
| SEGMENTATION | 0.174 | 0.389 |
| SENSORLESS | 0.075 | 0.078 |
| TICTACTOE | 0.000 | 0.023 |
| YEAST | 0.644 | 0.682 |

Table 3: Performance of Algorithm 1 compared to ERM on different datasets for linear models. We consider $\varepsilon = 1/m$ and $\varepsilon = 1/\sqrt{m}$, with $K = \alpha\sqrt{m}$ and $\alpha \in \{0, 0.4, 0.6, 0.8, 1\}$. We plot the empirical risk $\mathfrak{R}_{\mathcal{S}}(h)$ with its associated test risk $\mathfrak{R}_{\mu}(h)$.

(a) $K = 1$

| Dataset | Alg. 1 ($\frac{1}{m}$) | | Alg. 1 ($\frac{1}{\sqrt{m}}$) | |
|---|---|---|---|---|
| | $\mathfrak{R}_{\mathcal{S}}(h)$ | $\mathfrak{R}_{\mu}(h)$ | $\mathfrak{R}_{\mathcal{S}}(h)$ | $\mathfrak{R}_{\mu}(h)$ |
| ADULT | 0.207 | 0.207 | 0.248 | 0.248 |
| FASHIONMNIST | 0.142 | 0.155 | 0.126 | 0.149 |
| LETTER | 0.286 | 0.296 | 0.286 | 0.295 |
| MNIST | 0.067 | 0.092 | 0.069 | 0.094 |
| MUSHROOMS | 0.001 | 0.001 | 0.000 | 0.000 |
| NURSERY | 0.788 | 0.799 | 0.796 | 0.804 |
| PENDIGITS | 0.049 | 0.060 | 0.047 | 0.057 |
| PHISHING | 0.063 | 0.065 | 0.057 | 0.062 |
| SATIMAGE | 0.142 | 0.202 | 0.136 | 0.199 |
| SEGMENTATION | 0.053 | 0.151 | 0.079 | 0.176 |
| SENSORLESS | 0.907 | 0.911 | 0.907 | 0.911 |
| TICTACTOE | 0.013 | 0.021 | 0.013 | 0.021 |
| YEAST | 0.702 | 0.720 | 0.693 | 0.687 |

(b) $K = 0.4\sqrt{m}$

| Dataset | Alg. 1 ($\frac{1}{m}$) | | Alg. 1 ($\frac{1}{\sqrt{m}}$) | |
|---|---|---|---|---|
| | $\mathfrak{R}_{\mathcal{S}}(h)$ | $\mathfrak{R}_{\mu}(h)$ | $\mathfrak{R}_{\mathcal{S}}(h)$ | $\mathfrak{R}_{\mu}(h)$ |
| ADULT | 0.166 | 0.167 | 0.166 | 0.167 |
| FASHIONMNIST | 0.128 | 0.150 | 0.126 | 0.150 |
| LETTER | 0.285 | 0.296 | 0.286 | 0.297 |
| MNIST | 0.069 | 0.089 | 0.067 | 0.093 |
| MUSHROOMS | 0.001 | 0.001 | 0.001 | 0.001 |
| NURSERY | 0.760 | 0.778 | 0.769 | 0.781 |
| PENDIGITS | 0.050 | 0.061 | 0.048 | 0.061 |
| PHISHING | 0.062 | 0.067 | 0.065 | 0.068 |
| SATIMAGE | 0.565 | 0.773 | 0.137 | 0.200 |
| SEGMENTATION | 0.058 | 0.212 | 0.177 | 0.382 |
| SENSORLESS | 0.220 | 0.220 | 0.133 | 0.134 |
| TICTACTOE | 0.378 | 0.290 | 0.013 | 0.021 |
| YEAST | 0.488 | 0.478 | 0.492 | 0.478 |

(c) $K = 0.6\sqrt{m}$

| Dataset | Alg. 1 ($\frac{1}{m}$) | | Alg. 1 ($\frac{1}{\sqrt{m}}$) | |
|---|---|---|---|---|
| | $\mathfrak{R}_{\mathcal{S}}(h)$ | $\mathfrak{R}_{\mu}(h)$ | $\mathfrak{R}_{\mathcal{S}}(h)$ | $\mathfrak{R}_{\mu}(h)$ |
| ADULT | 0.166 | 0.167 | 0.166 | 0.167 |
| FASHIONMNIST | 0.127 | 0.147 | 0.127 | 0.150 |
| LETTER | 0.288 | 0.296 | 0.286 | 0.296 |
| MNIST | 0.067 | 0.092 | 0.067 | 0.093 |
| MUSHROOMS | 0.001 | 0.001 | 0.001 | 0.001 |
| NURSERY | 0.791 | 0.802 | 0.759 | 0.779 |
| PENDIGITS | 0.048 | 0.061 | 0.047 | 0.059 |
| PHISHING | 0.062 | 0.067 | 0.064 | 0.068 |
| SATIMAGE | 0.146 | 0.202 | 0.137 | 0.199 |
| SEGMENTATION | 0.058 | 0.215 | 0.058 | 0.204 |
| SENSORLESS | 0.129 | 0.130 | 0.130 | 0.130 |
| TICTACTOE | 0.013 | 0.021 | 0.013 | 0.021 |
| YEAST | 0.477 | 0.461 | 0.478 | 0.464 |

(d) $K = 0.8\sqrt{m}$

| Dataset | Alg. 1 ($\frac{1}{m}$) | | Alg. 1 ($\frac{1}{\sqrt{m}}$) | |
|---|---|---|---|---|
| | $\mathfrak{R}_{\mathcal{S}}(h)$ | $\mathfrak{R}_{\mu}(h)$ | $\mathfrak{R}_{\mathcal{S}}(h)$ | $\mathfrak{R}_{\mu}(h)$ |
| ADULT | 0.166 | 0.167 | 0.166 | 0.167 |
| FASHIONMNIST | 0.130 | 0.149 | 0.128 | 0.151 |
| LETTER | 0.285 | 0.296 | 0.288 | 0.297 |
| MNIST | 0.067 | 0.091 | 0.067 | 0.093 |
| MUSHROOMS | 0.001 | 0.001 | 0.001 | 0.001 |
| NURSERY | 0.771 | 0.787 | 0.758 | 0.778 |
| PENDIGITS | 0.047 | 0.060 | 0.047 | 0.059 |
| PHISHING | 0.062 | 0.066 | 0.065 | 0.068 |
| SATIMAGE | 0.168 | 0.216 | 0.137 | 0.199 |
| SEGMENTATION | 0.053 | 0.212 | 0.052 | 0.204 |
| SENSORLESS | 0.129 | 0.130 | 0.132 | 0.132 |
| TICTACTOE | 0.013 | 0.021 | 0.013 | 0.021 |
| YEAST | 0.476 | 0.461 | 0.477 | 0.460 |

(e) $K = \sqrt{m}$

| Dataset | Alg. 1 ($\frac{1}{m}$) | | Alg. 1 ($\frac{1}{\sqrt{m}}$) | |
|---|---|---|---|---|
| | $\mathfrak{R}_{\mathcal{S}}(h)$ | $\mathfrak{R}_{\mu}(h)$ | $\mathfrak{R}_{\mathcal{S}}(h)$ | $\mathfrak{R}_{\mu}(h)$ |
| ADULT | 0.166 | 0.167 | 0.166 | 0.167 |
| FASHIONMNIST | 0.354 | 0.361 | 0.127 | 0.151 |
| LETTER | 0.287 | 0.296 | 0.288 | 0.298 |
| MNIST | 0.068 | 0.092 | 0.065 | 0.092 |
| MUSHROOMS | 0.001 | 0.001 | 0.001 | 0.001 |
| NURSERY | 0.795 | 0.805 | 0.796 | 0.805 |
| PENDIGITS | 0.050 | 0.062 | 0.047 | 0.059 |
| PHISHING | 0.062 | 0.067 | 0.065 | 0.067 |
| SATIMAGE | 0.143 | 0.200 | 0.137 | 0.201 |
| SEGMENTATION | 0.055 | 0.210 | 0.055 | 0.212 |
| SENSORLESS | 0.130 | 0.130 | 0.131 | 0.132 |
| TICTACTOE | 0.013 | 0.021 | 0.392 | 0.301 |
| YEAST | 0.476 | 0.456 | 0.476 | 0.457 |

(f) ERM

| Dataset | $\mathfrak{R}_{\mathcal{S}}(h)$ | $\mathfrak{R}_{\mu}(h)$ |
|---|---|---|
| ADULT | 0.166 | 0.167 |
| FASHIONMNIST | 0.139 | 0.153 |
| LETTER | 0.287 | 0.297 |
| MNIST | 0.065 | 0.091 |
| MUSHROOMS | 0.001 | 0.001 |
| NURSERY | 0.794 | 0.807 |
| PENDIGITS | 0.052 | 0.064 |
| PHISHING | 0.064 | 0.067 |
| SATIMAGE | 0.148 | 0.209 |
| SEGMENTATION | 0.087 | 0.232 |
| SENSORLESS | 0.134 | 0.136 |
| TICTACTOE | 0.228 | 0.238 |
| YEAST | 0.470 | 0.427 |

Table 4: Performance of ERM with weight decay (with the L2 regularisation) for linear and neural network models.

(a) Linear

| Dataset | L2 Reg. ($\frac{1}{m}$) | | L2 Reg. ($\frac{1}{\sqrt{m}}$) | |
|---|---|---|---|---|
| | $\mathfrak{R}_{\mathcal{S}}(h)$ | $\mathfrak{R}_{\mu}(h)$ | $\mathfrak{R}_{\mathcal{S}}(h)$ | $\mathfrak{R}_{\mu}(h)$ |
| ADULT | 0.207 | 0.207 | 0.248 | 0.248 |
| FASHIONMNIST | 0.141 | 0.149 | 0.127 | 0.150 |
| LETTER | 0.285 | 0.295 | 0.285 | 0.296 |
| MNIST | 0.067 | 0.092 | 0.066 | 0.092 |
| MUSHROOMS | 0.001 | 0.001 | 0.000 | 0.000 |
| NURSERY | 0.788 | 0.799 | 0.796 | 0.804 |
| PENDIGITS | 0.049 | 0.060 | 0.047 | 0.057 |
| PHISHING | 0.063 | 0.065 | 0.057 | 0.062 |
| SATIMAGE | 0.144 | 0.203 | 0.138 | 0.200 |
| SEGMENTATION | 0.058 | 0.157 | 0.075 | 0.177 |
| SENSORLESS | 0.907 | 0.911 | 0.907 | 0.911 |
| TICTACTOE | 0.013 | 0.021 | 0.013 | 0.021 |
| YEAST | 0.702 | 0.720 | 0.693 | 0.687 |

(b) NN

| Dataset | L2 Reg. ($\frac{1}{m}$) | | L2 Reg. ($\frac{1}{\sqrt{m}}$) | |
|---|---|---|---|---|
| | $\mathfrak{R}_{\mathcal{S}}(h)$ | $\mathfrak{R}_{\mu}(h)$ | $\mathfrak{R}_{\mathcal{S}}(h)$ | $\mathfrak{R}_{\mu}(h)$ |
| ADULT | 0.207 | 0.207 | 0.248 | 0.248 |
| FASHIONMNIST | 0.160 | 0.166 | 0.159 | 0.164 |
| LETTER | 0.261 | 0.275 | 0.256 | 0.269 |
| MNIST | 0.116 | 0.125 | 0.084 | 0.095 |
| MUSHROOMS | 0.000 | 0.000 | 0.000 | 0.000 |
| NURSERY | 0.704 | 0.721 | 0.770 | 0.788 |
| PENDIGITS | 0.009 | 0.022 | 0.012 | 0.026 |
| PHISHING | 0.042 | 0.050 | 0.054 | 0.059 |
| SATIMAGE | 0.150 | 0.215 | 0.143 | 0.205 |
| SEGMENTATION | 0.141 | 0.216 | 0.198 | 0.371 |
| SENSORLESS | 0.907 | 0.911 | 0.907 | 0.911 |
| TICTACTOE | 0.000 | 0.046 | 0.000 | 0.021 |
| YEAST | 0.662 | 0.674 | 0.693 | 0.683 |

