# OpenReview forum: "Learning via Wasserstein-Based High Probability Generalisation Bounds"
_NeurIPS.cc/2023/Conference — NeurIPS 2023 poster_

### Official Review · Reviewer_jPqk · 2023-07-05

**Soundness:** 3 good
**Presentation:** 4 excellent
**Contribution:** 4 excellent
**Rating:** 7
**Confidence:** 4

**Summary:**

This paper focuses on studying PAC-Bayes generalization bounds based on the Wasserstein distance. It introduces novel high-probability bounds applicable to both batch learning (i.i.d. setting) and online learning (non-i.i.d. setting). Unlike previous PAC-Bayes bounds, their bounds are not limited to bounded or sub-Gaussian losses. Furthermore, leveraging their theoretical findings, the authors propose a new SRM training strategy and provide empirical evidence to validate its effectiveness.

**Strengths:**

Compared to previous results, the Wasserstein-distance PAC-Bayes bounds presented in this paper require fewer or weaker assumptions. Additionally, unlike previous PAC-Bayes bounds that typically invoke Gaussian distributions, their bounds accommodate discrete Dirac distributions for both the prior and posterior distributions.

**Weaknesses:**

Regarding the Wasserstein regularization part, it might be necessary to empirically compare it with similar styles of regularization, such as weight decay $||w||$ and the 'distance to initialization' $||w-w_0||$. Although I believe that a data-dependent prior will provide a tighter bound, it is uncertain whether the data-dependent prior can significantly outperform a data-independent prior (e.g., $0, w_0$) considering the significantly increased computational time required.

**Questions:**

Major concerns:

1. There might be a small error in Theorem 2: in the proof of Theorem 2 (Section B.2), you let $\lambda_i=\sqrt{\frac{\ln{(K/\delta)}}{|S_i|}}$ in the last step (i.e. Line 636). However, this choice does not yield the desired $\sum_{i=1}^K\sqrt{\frac{|S_i|\ln{(K/\delta)}}{m^2}}$ in your bound. If I understand correctly, you could consider setting $\lambda_i=\sqrt{\frac{2\ln{(K/\delta)}}{|S_i|}}$, which would result in the last term in the bound becoming $\sum_{i=1}^K\sqrt{\frac{2|S_i|\ln{(K/\delta)}}{m^2}}$. This may be the smallest achievable second term in Section B.2.

2. In Line 206-207, you argue that Eq. (3) contains a trade-off parameter $\lambda$, while Eq. (2) does not. However, it is worth noting that a similar parameter, $\epsilon$, is also involved in your practical training objective, namely Eq. (5). Therefore, the argument regarding the presence of $\lambda$ may not be entirely fair. Perhaps removing Line 206-207 could be considered?

3. Have you considered incorporating the empirical variance term into your algorithm? Since you emphasize its reliance on the prior rather than the posterior, I would appreciate it if you could demonstrate some practical implications.

4. Under the same conditions, such as bounded loss or even zero-one loss, is it possible to compare your Wasserstein-based PAC-Bayes bounds with the standard KL-based PAC-Bayes bounds, especially those that also include variance terms as mentioned in [1,2,3]?

Minor comments:

1. Line 131: there is a missing period after $S_K$.

2. RHS of the equation on Line 633: $L{|S_i|}\Longrightarrow L_{|S_i|}$.

3. RHS of the equation on Line 635: $\lambda [S_i|]\Longrightarrow \lambda |S_i|$.

[1] Seldin, Yevgeny, et al. "PAC-Bayesian inequalities for martingales." IEEE Transactions on Information Theory 58.12 (2012): 7086-7093.

[2] Tolstikhin, Ilya O., and Yevgeny Seldin. "PAC-Bayes-empirical-Bernstein inequality." NeurIPS 2013.

[3] Mhammedi, Zakaria, Peter Grünwald, and Benjamin Guedj. "PAC-Bayes un-expected Bernstein inequality." NeurIPS 2019.

**Limitations:**

There is no potential negative societal impact.

---

> ### Author Rebuttal · Authors · 2023-08-09
>
> We thank you for your careful review, and we answer your concerns below.
>
> **Weaknesses**
>
> As noted in the general answer, the 'distance to initialisation' is a particular case of our Eq. (5) when $K=1$. This particular case has been plotted in Table 3 of Appendix C.4. More precisely, this table shows that, on most of the datasets, considering data-dependent priors leads to sharper results. This shows the efficiency of our method compared to the 'distance to initialisation' regularisation. Furthermore, we ran a novel experiment with the weight decay generalisation which is gathered in the pdf document updated with the rebuttal. This experiment shows concretely that on a few datasets (namely SENSORLESS, YEAST), when our predictors are neural networks, the weight decay regularisation fails to learn while our method succeeds, as shown in Tables 1 (of our work) and 4 (of the new pdf). More generally, weight decay regularisation outperforms our method 3 times over 11 for batch learning with neural nets.
>
> We hope the experiments and the analysis address your concerns.
>
> **Major concerns**
> 1. Thank you for spotting this, you are actually right. Our choice of $\lambda$ was not correct, while yours is. We will correct it in the next version of the document.
>
> 2. Thank you again; this remark is indeed unfair. After looking at our practical optimisation procedure, we will remove those two lines.
>
> 3. According to our theoretical bounds, it is legitimate to incorporate the empirical variance term when considering a heavy-tailed loss that takes value in all $\mathbb{R}$. This situation is in line, for instance, with reinforcement learning with heavy-tailed rewards, as noticed in line 155. However, we proved in Theorems 2 and 4 that if the loss is nonnegative, then the empirical variance does not intervene in the bound. Thus, as our optimisation goal is driven by the theory, it does not seem relevant to us to incorporate such an empirical variance in a classification problem. However, in order to provide a concrete situation where order 2 moments on the prior only have a positive influence (rather than on the pair prior/posterior) we propose the following example.
>
> Assume that order 2 moments of the prior *and all considered posteriors* have to be uniformly bounded by a certain $C$.
> Let the data distribution of a point $z$ to admit only a finite variance and consider the loss $\ell(h,z)= |h-z|$ where both $h,z$ lie in the real axis. Then, assuming $\mathbb{E}[z^2]\le V$, then by Cauchy Schwarz, we have  $\mathbb{E}[\ell(h,z)^2] \le \mathbb{E}[h^2] + 2V\mathbb{E}[|h|] +V^2 \le C$, the last inequality being our assumption. Then, this invites us to consider posterior distributions with first and second moments satisfying the previous inequality, which is restrictive. On the contrary, assuming only such an assumption on the prior distribution, we are allowed to move freely on the space of distribution to find the best posterior.
> We thank you for asking for more details, as this short example will improve the next version of our manuscript.
>
> 4. We provide below further elements of comparison between [1,2,3] and our results, beyond the fact that we used the Wasserstein distance instead of KL divergence.
>
> The bounds of [1] are PAC-Bayes bounds for martingales, we focus on their Theorem 4 which controls an average of martingales, similarly to our Thm 1. Under a boundedness assumption, they recover a McAllester-typed bound while Thm 1 is more of a Catoni-typed result. Also, their Theorem 7 is a Catoni-typed bound involving a conditional variance, similar to our Theorem 4. However, they require to bound uniformly the variance on all the predictor sets, which is more restrictive than the assumption of bounding the averaged variance with respect to priors, which is what we required to perform Theorem 4.
>
> The bounds of [2] have been obtained for bounded loss functions and iid data; this allows us to exploit techniques involving the 'small' kl divergence between two Bernoulli r.v. Thus, their bounds have been obtained through a certain lower bound on this kl term. Their main result (Theorem 4) contains a term of variance times a KL divergence, exhibiting an interaction between those two quantities. On the contrary, our bounds decoupled the influence of the empirical variance from the one from the complexity term (i.e., the Wasserstein distance). The major advantage to this is that we do not have to assume a constraint on our choice of posterior, such as the assumption below Eq. (7) in [2]. This allows our learning algorithms to move freely in the space of distributions to find our posterior.
>
> The bounds of [3] are a continuation of [2] as they provide tighter PAC-Bayes Bernstein inequalities. However, we notice that the involved technical toolbox is closer to ours as they widely used Exponential Stochastic Inequalities (ESI). Indeed, our supermartingale toolbox can be seen as a carefully designed ESI. However, their results remain valid for bounded losses and i.i.d data, which remains more restrictive than our assumptions.
>
> We hope this clarifies your concerns and are happy to respond to any additional questions.

---

> > ### Comment · Reviewer_jPqk · 2023-08-11
> > **Thanks**
> >
> > Thank you for your response! All my questions are clarified.

---

> > > ### Author Response · Authors · 2023-08-17
> > >
> > > Thank you for acknowledging that your concerns are clarified and for your review.

---

### Official Review · Reviewer_mMbV · 2023-07-05

**Soundness:** 3 good
**Presentation:** 3 good
**Contribution:** 3 good
**Rating:** 8
**Confidence:** 3

**Summary:**

This paper introduces Wasserstein-based PAC-Bayes bounds both for offline (batch learning) and online learning.

These bounds are obtained with a clever use of the Kantorovich-Rubinstein duality together with a control of some additional terms appearing in the proof that only depend on the prior. To use the Kantorovich-Rubinstein duality they require the loss to be Lipschitz and to control the additional terms they require the loss to have a bounded second moment, following the techniques from [HG23a]. The extension from offline to online learning comes almost "for free" in the analysis due to the super-martingale strategies developed in [HG23a]. This way, the presented bounds have a set of weak assumptions (Lipschitzess + bounded second moment) which makes them amenable to losses with potentially heavy tails.

Another aspect of the bounds is that they allow data-dependent priors. For this, they separate the training dataset $\mathcal{S}$ into $K$ subsets $\mathcal{S}\_i$and dedicate $K$ data-dependent priors $\pi\_i$ that depend on $\mathcal{S} \setminus \mathcal{S}\_i$. This way, they have an average of Wasserstein distances of the posterior and each of these priors weighted accordingly. This way, for a large number of samples the posterior and the priors start to align helping the bound to become tighter.

Finally, they employ their bounds to derive regularization terms that are employed to design algorithms both in offline and online learning. They showcase how the usage of these algorithms does not decrease the performance of linear and deep models and that, in some cases, they help these models achieve better performance.







**Strengths:**

The strengths of this paper are multiple, but the most important are:
- The presented theorems are general (Lipschitz and second-moment assumptions only).
- The proofs of the presented theorems are exposed in a simple way, even though the final results are useful.
- The theorems allow for deterministic predictors, which is one of the biggest drawbacks of PAC-Bayes bounds.
- The results are validated empirically.

**Weaknesses:**

There are not many major issues with the paper. My biggest concern is the following:
- In the section "Batch algorithm" starting in line 275, you use the equation (5) to guide the training of the posterior, which includes the distance $\lVert \textbf{w} - \textbf{w}_i \rVert_2$ that comes from the Wasserstein distance. However, the employed loss is only Lipschitz with respect to the outputs. How do you justify then the usage of the $\ell$_2 norm of the weights, as it could be that the loss is not Lipschitz with respect to the weights? The same happens later in equations (6) and (7).

Other smaller issues are the following:
- The experimental results don't seem too strong compared with previous methods such as the one presented in [PORPH+21]. Although they use probabilistic networks, it would have been interesting to observe how the two methods compare to each other with the same network architecture. For instance, with a FCN and Gaussian priors, they show population risk certificates of ~0.02, while the presented here are of ~0.09 and ~0.12.
- Some of the writing can be clarified. In particular, I am referring to sentences of the type: "for all $i$, $\mathcal{S}$,$\pi\_i(S,\cdot)$ is $\mathcal{F}\_{i-1}$ measurable". This is not very clear, I would recommend changing all these sentences for something like "the distribution $\pi\_i(S,\cdot)$ is $\mathcal{F}\_{i-1}$ measurable for all $i$ and all $\mathcal{S}$". This small change can improve readability.
- I missed some comments on Proposition 5 of [AEM22], which is a Seeger-type bound. How do the presented bounds compare to that?
- I am not sure the implementation of OGD is correct, given that it ends up having a larger empirical risk than the regularized version. Maybe I am missing something here, or maybe this showcased some limitations of OGD I was not aware of. I also mentioned it explicitly as a question below.

Some typos:
- In line 29, it should be $R\_{\mu}(h) - \hat{R}\_{\mathcal{S}}(h)$.
- In lines 192 and 193, you forgot to include the hat in $\hat{R}$.
- In line 354, it should be $\mathfrak{C}\_{\mathcal{S}}$.

**Questions:**

* How do you justify the usage of the $\ell_2$ norm on the weights in equations (5), (6), and (7) if the employed loss is only Lipschitz with respect to the outputs of the hypotheses $h_{\textbf{w}}$ but not necessarily with respect to the weights themselves?
* What is the intuition for the fact that Algorithm 2 has a smaller empirical risk than OGD? It seems weird given the fact that Algorithm 2 is essentially a regularized version of OGD, which is an algorithm for ERM.

**Limitations:**

Some of the limitations are mentioned throughout the text, yes.

---

> ### Author Rebuttal · Authors · 2023-08-09
>
> Thank you for the positive review and for pointing out the 'multiple strengths'.
>
> **Your major concern**
>
> Thank you for spotting this - see also our general response. A Wasserstein is defined on a Polish space w.r.t a distance $d$. Indeed our Algos in Eqs. (5-7) must be described w.r.t $d$ and not an Euclidean norm on the weights, we fixed this in the revision. When the predictor space $\cal{H}$ is parametrised by a Euclidean space with Euclidean distance $||.||$, the distance between two predictors $h\_{\bf w},h\_{{\bf w}'}\in \cal{H}^2$ is $d(h\_{\bf w},h\_{{\bf w}'})=||{\bf w}-{\bf w}'||$. This raises the question of transferring the Lipschitzness on $h_{\bf w}$ to one on ${\bf w}$. Lemma 8 ensures this is true for the linear model, but we omitted to prove it was true for our neural nets when weights are bounded. We prove below a new lemma showing that with clipped weights in a Fully Connected Network (FCN) with Lipschitz activation functions, then our loss is Lipschitz w.r.t the weights and we can use the $\ell_2$ norm in our objective. Note that the clipping constant plays no role so we set it to a large value to avoid interfering with optimisation. This has been added to Sec 4.1 and we hope this addresses your concern.
>
> **Minor**
> 1. Discrepancy in performance: this is because we do not minimise the same objective (they use the cross-entropy loss).
> 2. Thanks - added to the revised paper.
> 3. We do not compare to Prop. 5 of [AEM22] but to their Thms 11-12, which are versions of Prop. 5 for Wasserstein. See paragraph starting on l196. Our proof does not allow the use of the 'small' kl divergence as we need $\sum_{i=1}^m R_{\mu}(h)-\ell(h,{\bf z}_i)$ to apply our supermartingale trick. Note that our result holds with weaker assumptions than Thms 11-12.
>
> **Questions**
> 1. The L2 norm in experiments is motivated by Lemma 8 and the new lemma. In both cases we prove the loss is Lipschitz w.r.t the output $h_{{\bf w}}$ and ${\bf w}$.
>
> 2. For each datum, OGD is performing only one gradient descent step while our method solves a (constrained) optimization problem. Hence, we believe that OGD might suffer from underfitting as one gradient step at each time may not be enough for a non-regularised method. Since Rev. tL5n has the same concern, we decided to add the answer to the paper after line 355.
>
> **Lemma's statement**
>
> We define recursively a FCN as follows: for a vector ${\bf W}\_1= \text{vect}(\{W\_1,b\})$ and an input datum $x$, $FCN\_1({\bf W}\_1,x)=\sigma\_1(W\_1x+b\_1)$, where $\sigma\_1$ is the activation function. Also, for any $i\geq 2$ we define for a vector ${\bf W}\_i=(W\_i,b\_i,{\bf W}\_{i-1})$ (defined recursively as well), $FCN\_i({\bf W}\_i,x)=\sigma\_i(W\_iFCN\_{i-1}({\bf W}\_{i-1},x)+b\_i)$.
> Then, setting ${\bf z}=(x,y)$ a datum and  $h\_i(x):=FCN\_i({\bf W}\_{i},x)$ we can rewrite our loss as a function of $({\bf W}\_i,{\bf z})$.
>
> **Lemma.** *Assume that all weight matrices of ${\bf W}\_i$ are bounded and that the activation functions are Lipschitz with constant bounded by $K\_\sigma$. Then for any datum ${\bf z}=(x,y)$, any $i,{\bf W}\_i\rightarrow\ell({\bf W\_i},{\bf z})$ is Lipschitz.*
>
> *Proof.* We consider the Frobenius norm on matrices as ${\bf W}_2$ is a vector as we consider the L2-norm on the vector. We prove the result for $i=2$, assuming it is true for $i=1$. We then explain how this proof generalises the case $i=1$ and works recursively. Let ${\bf z},{\bf W}_2,{\bf W}_2'$, for clarity we write $FCN_2(x):=FCN({\bf W}_2,x)$ and $FCN_2'(x):=FCN({\bf W}_2',x)$.  As $\ell$ is $\eta$-Lipschitz on the outputs $FCN_2(x), FCN_2'(x)$:
>
> $$|\ell({\bf W\_2},{\bf z})-\ell({\bf W'\_2},{\bf z})|$$
> $$\le\eta||FCN\_2(x)-FCN\_2'(x)||$$
> $$\le\eta||\sigma\_2(W\_2FCN\_{1}(x)+b\_2)-\sigma\_2(W\_2'FCN\_{1}'(x)+b\_2')||$$
> $$\le\eta K\_\sigma||W\_2FCN\_{1}(x)+b\_2-W\_2'FCN\_{1}'(x)-b\_2'||$$
> $$ \le\eta K\_\sigma( ||(W\_2- W\_2')FCN\_{1}(x)||+||W\_2'(FCN\_{1}(x)-\text{FCN\}_{1}'(x))||+||b\_2-b\_2'||)$$
>
> Then we have $||(W\_2-W\_2')FCN\_1(x)||\le||(W\_2-W\_2')||\_F||FCN\_1(x)|| \le  K\_x ||(W\_2- W\_2')||\_F$. The second inequality holding as $FCN\_1(x)$ is a continuous function of the weights. Indeed, as on a compact space, a continuous function reaches its maximum, then its norm is bounded by a certain $K\_x$. Also, as the weights are bounded, any weight matrix has its norm bounded by a certain $K\_{W}$ thus $||W\_2'(FCN\_1(x)FCN\_1'(x)|| \le ||W\_2'||\_F||(FCN\_1(x)-FCN\_1'(x)||\le K\_{W}||FCN\_1(x)-FCN\_1'(x)||$. Finally, taking $K\_{temp}=\eta K\_\sigma\max(K\_{x},K\_W,1)$ gives: $$|\ell({\bf W\_2},{\bf z})-\ell({\bf W'\_2},{\bf z})|\le K\_{temp}(||(W\_2-W\_2')||\_F+||b\_2-b\_2'||+||FCN\_1(x)-FCN\_1'(x)||).$$
>
> Exploiting the recursive assumption that $FCN\_1$ is Lipschitz with respect to its weights ${\bf W}\_1$ gives  $||FCN\_1(x)-FCN\_1'(x)||\le K\_1||{\bf W}\_1-{\bf W}\_1'||$. If we denote by $(W_2,b_2)$ the vector of all concatenated weights, notice that $||(W\_2- W\_2')||\_F+||b\_2-b\_2'||=\sqrt{(||(W\_2-W\_2')||\_F+||b\_2-b\_2'||)^2}$$\le\sqrt{2(||(W\_2-W\_2')||\_F^2+||b\_2-b\_2'||^2)}=\sqrt{2}||(W_2,b_2)-(W_2',b_2')||$ (we used that for any real numbers $a,b,(a+b)^2\le 2(a^2+b^2)$). We then have: $$|\ell({\bf W\_2}, {\bf z})-\ell({\bf W'\_2},{\bf z})|$$
> $$\le K\_{temp}\max(\sqrt{2},K\_1)( ||(W\_2,b\_2)-(W\_2',b\_2')||+||{\bf W}\_1-{\bf W}\_1'||)$$
> $$\le\sqrt{2}K_{temp}\max(\sqrt{2},K_1)||{\bf W}_2-{\bf W}_2'||.$$
> The last line holds by reusing the same calculation trick. This concludes the proof for $i=2$. For $i=1$, the same proof holds by replacing $W_2,b_2,FCN_2$ by $W_1,b_1,FCN_1$ and replacing $FCN_1(x), FCN_1'(x)$ by $x$ (we then do not need to assume a recursive Lipschitz behaviour). Then the result holds for $i=1$.
>
> We then apply a recursive argument at rank $i$ by assuming the result at rank $i-1$ reusing the same proof by replacing $W\_2,b\_2,FCN\_2$ by $W\_i,b\_i,FCN\_i$ and $FCN\_1(x),FCN\_1'(x)$ by $FCN\_{i-1}(x),FCN\_{i-1}'(x)$. This concludes the proof.

---

> > ### Comment · Reviewer_mMbV · 2023-08-14
> > **Answer to rebuttal**
> >
> > Thank you for your rebuttal.
> >
> > My major concern is now addressed with the new Lemma, thanks. This only begs the question on how to choose the hyper-parameter $\varepsilon$ with respect to that Lipschitz constant or, in other words, how much the bound is used for regularization. But you already discuss that in the paragraph after (5), so a small further discussion will do.
> >
> > Also, all my minor concerns and questions are clarified.
> >
> > (P.S.: I missed that in [AEM22, Prop. 5] they needed that $m \cdot \Delta\_S^{\mathrm{kl}} \in \mathcal{F}\_S$ for some reason. Then, I thought one could choose $\mathcal{F}\_S$ to be the set of Lipschitz functions. Sorry for that.).

---

> > > ### Author Response · Authors · 2023-08-17
> > >
> > > Thank you for answering to the rebuttal and your review.

---

> > > > ### Comment · Reviewer_mMbV · 2023-08-18
> > > > **Score update**
> > > >
> > > > Dear authors,
> > > >
> > > > We can now update the scores. Since my main concern is addressed, I increased it by 2 points.

---

> > > > > ### Author Response · Authors · 2023-08-18
> > > > >
> > > > > Dear Reviewer mMbV,
> > > > >
> > > > > We warmly thank you for both your constructive review which strengthen our manuscript, and the increased score.

---

### Official Review · Reviewer_tL5n · 2023-07-05

**Soundness:** 4 excellent
**Presentation:** 4 excellent
**Contribution:** 4 excellent
**Rating:** 8
**Confidence:** 3

**Summary:**

This work propose PAC-Bayesian learning with the KL divergence replaced by the Wasserstein distance on the metric space of hypotheses $(\mathcal{H}, d)$. Denoting the data distribution by $\mathcal{D}$, the authors study the following learning problems:

1. Batch learning. Under the assumption that the loss function has bounded 2nd moment with respect to $\mathcal{D}$ and priors over $\mathcal{H}$, the authors derive a generalization bound in terms of $LW$, where $W$ is the Wasserstein distance between posterior and prior distributions over $\mathcal{H}$ and $L$ is the Lipschitz constant of the loss function (with respect to $d$).
2. Online learning. Under the assumption that the loss function has bounded 2nd _conditional_ moment with respect to $\mathcal{D}$ and priors over $\mathcal{H}$, the authors derive a similar generalization bound as above, where the population risk is defined by the sum of conditional expections of the loss, given a filtration adapted to the sequence of data.

Both bounds can be applied to heavy-tailed loss functions with bounded 2nd moment. And they can be used to derive new learning problems regularized by $LW$. Experimental results with a linear model and an NN model show that these algorithms have better test risks than ERM and OGD on several dataset, among which online learning with the NN model shows the most improvement.

**Strengths:**

- This is a strong result in the field of PAC-Bayes learning. Replacing the KL divergence by the Wasserstein distance is logical when the geometry of the underlying metric space is in consideration.
- The bound is just as nice as the KL divergence version of the PAC-Bayes bound. And the ability to convert the bound into the algorithm is powerful.
- The experimental results show that the bound can be used to improve models' generalization.


**Weaknesses:**

- It seems that the batch size substantially affects the test risks. For example,  learning with $\epsilon = 1/m$ on MNIST and TICTACTOE with the linear model, and TICTACTOE on the NN model. Are there any specific reasons that make these models underperform on these datasets?
- Alg 2. seemingly performs as well as OGD with the linear model; on the other hand, the former does noticeably better with the NN model. The authors might want to discuss on why the NN model performs well in this case.
- A minor comment on the organization: In Section 4.1: Batch algorithm, the authors set $L=1/2$ and proceed to define a specific loss that has the said Lipschitz constant, which is quite confusing to read. I think it is more logical to introduce the loss function first.

**Questions:**

- Though not required, but I think high level ideas of the proofs of Theorem 1 and 3 would be a nice addition.
-  In Section 4.1: Batch algorithm, has the definition of $d(h,h_i)$ been defined before? I guess the authors have make a convention that $d(h,h_i)=d(w,w_i)$.
- Can the authors performs the experiment against ERM with standard regularization as well?

**Limitations:**

The authors might want to add some limitation of the bounds. For example, it might be difficult to compute the Wasserstein distance over nonparametric models, such as kNN.

---

> ### Author Rebuttal · Authors · 2023-08-09
>
> We thank you for your enthusiastic review. We are thrilled to read that you see this work as a 'strong result in the field of PAC-Bayes learning'. We answer your questions below.
>
> **About the batch size**
>
> This is indeed a fair question, and unfortunately at this stage we found no particular reason why $\varepsilon$ affects the test risks. For a given dataset, there is no obvious choice for the model and the parameters $\varepsilon$ and $K$. For instance, in Table 3 of Appendix C, for the SEGMENTATION dataset, the parameters $K=1,\varepsilon=\frac{1}{m}$ are optimal (in terms of test risks) for both models. As $K=1$ means that our single prior is data-free, this shows that the intrinsic structure of SEGMENTATION makes it less sensitive to both the information contained in the prior ($K=1$ meaning data-free prior) and the place of the prior itself ($\varepsilon=1/m$ meaning that we give less weight to the regularisation within our optimisation procedure). On the contrary, in Table 1, the YEAST dataset performs significantly better when $\varepsilon=1/\sqrt{m}$ (and $K=0.2\sqrt{m}$), exhibiting a positive impact of our data-dependent priors. We will add a discussion on this matter in the next version.
>
> **Performance of Alg.2**
>
> As stated in lines 353 to 355, we believe that OGD can suffer from underfitting. Indeed, only one gradient descent step is done for each new datum $(\mathbf{x}\_i, y\_i)$, which might not be sufficient to decrease the loss. Instead, our method solves the problem associated with Eq. (7) and constrains the descent with the norm $||\mathbf{w}-\mathbf{w}\_{i-1}||$. In the linear case, we are able to solve the problem, but for the neural network case, we minimise the approximation for 10 steps. As the question was also raised by reviewer mMbV, this answer will be added in the next revision of the paper after line 355.
>
> **About the organisation of Sec. 4.1**
>
>  Thank you for notifying this; we acknowledge in our general answer that the beginning of Sec. 4.1 would benefit of more clarity. More precisely, we will move the second paragraph of the 'Batch algorithm' part before Eq. (5). We will also define properly $d$ on $\mathcal{H}$ to make explicit the convention $d(h\_{\mathbf{w}},h\_{\mathbf{w}'})=||\mathbf{w}-\mathbf{w}'||$ and will replace the Euclidean norm by the distance $d$ in Eqs. (5), (6), (7).
>
> **High-level ideas of the proof**
>
> Due to space constraints, we did not extend the theoretical discussion as much as we wanted. However, we are happy to put more discussion about both the high-level ideas of the proof and our set of assumptions in the next version.
>
> **Additional experiments**
>
> Yes, we will provide additional experiments. As written in the general answer, we implemented in Table 3 of Appendix C.4 the particular case $K=1$ of Algorithm 1, which corresponds to the classical 'distance to initialisation' regularisation (i.e., $||\mathbf{w}-\mathbf{w}\_0||$ with $\mathbf{w}\_0$ being the initialisation. It shows that in most cases, taking $K$ greater than 1 (i.e., fully exploiting the theoretical ground offered by our PAC-Bayes bounds) leads to better results in a vast majority of datasets. Furthermore, as it was asked by Rev. jPqk to run an ERM with the 'weight decay' regularisation (i.e, $||\mathbf{w}||$), this has been added in the pdf document linked to the general answer. This experiment shows concretely that on a few datasets (namely sensorless and yeast) when our predictors are neural nets, the weight decay regularisation fails to learn while ours succeeds as shown in Tables 1 (of our work) and 4 (of the new pdf). More generally, when considering neural nets, the weight decay regularisation outperforms our method 3 times over 11 for batch learning.
>
> Again, we thank you for your review and hope our answers address all of your concerns.

---

> > ### Comment · Reviewer_tL5n · 2023-08-17
> > **Thank you**
> >
> > I thank the authors for addressing all of my concerns. There is an open problem of finding a suitable batch size left which is interesting in its own right.

---

> > > ### Author Response · Authors · 2023-08-17
> > >
> > > Thank you for your reply and again for your (positive) review. Indeed, the choice of the batch size has been made empirically in our paper. This suggests an important open question which we are happy to mention in the main document and to investigate in future works. Again, we thank you for your time.

---

### Official Review · Reviewer_gr5G · 2023-07-06

**Soundness:** 3 good
**Presentation:** 3 good
**Contribution:** 4 excellent
**Rating:** 6
**Confidence:** 3

**Summary:**

The paper builds on recent advances in PAC-Bayesian learning and derives new Wasserstein distance-based generalization bounds. Besides batch learning for iid data, a first set of results are derived for online learning (with non iid data). The authors also provide tight bounds for the case of heavy tailed loss with bounded second order moments. Through experimental results the authors argue that these bounds can inspire new learning algorithms that achieve comparable (and in some cases better) empirical performance than classical benchmarks (ERM for batch and OGD for online setting).


**Strengths:**


The introduction and main results are well placed in the context of recent literature in this area. Although this reviewer has not worked on the topic of PAC-Bayes learning, it was possible to understand the authors’ contribution relative to work by Haddouche and Guedj, Catoni and co-authors, and recent paper of Amit et al from NeurIPS 2022. One would have expected more clarity on the use of “supermatingale toolbox” in the main text of the paper, but I suppose this can be addressed using minor revisions of the paper.



**Weaknesses:**


However, there are several points that impede the broader understanding of the work. Firstly, the role and possible limitations imposed by L-Lipschitz assumption should be clarified.

In the discussion of Theorem 1, the authors mention that the variance terms are considered with respect to the prior distributions $\pi_{i,\mathcal{S}}$ and not $\rho$. Can you clearly explain why this is a strength of the result (if so)?

In comment after equation (2), I agree that Wasserstein distance as regularizer…allows the user of multiple priors. But the issue of selecting K (as opposed to choice of $\lambda$ in classical algorithm). The authors do provide some discussion of role of $K$ but the nuance of its role in regularization is still not very clear (besides, the discussion on tradeoff needs a bit more clarity).

Appreciate the interest in online setting, but the contribution/merits of algorithm based on equation (4) are not very clear. Firstly, what is the conceptual novelty here in comparison to online counterpart to equation (3)? Secondly, the particularization to Dirac masses seems like a restriction. The algorithmic aspects need a bit more clarity as well.

The computational results seem promising and an assertive discussion on comparison with ERM and OGD benchmarks is needed.


**Questions:**

Please see my comments/suggestion in the Weaknesses section.

**Limitations:**

Would have liked to see a more robust discussion on the limitations of the proposed approach. A clear explanation about when and to what extent geometry of data space can improve the results obtained through Wasserstein distance based approach would be much appreciated. Also comment on future work to refine results on online setting.

---

> ### Author Rebuttal · Authors · 2023-08-09
>
> We thank you for your thoughtful review. We are encouraged to read that 'the introduction and main results are well-placed in the context of recent literature' and that you appreciated the goal of our work, even though 'this reviewer has not worked on the topic of PAC-Bayes learning', as one of our goals is to reach a broader audience.
>
> **Role of Lipschitz assumption**
>
> As the Wasserstein distance is used in PAC-Bayes as a geometric notion of complexity, the theoretical role of the Lipschitz assumption is to allow the use of such a distance. On the contrary, the KL divergence comes from information theory and does not require such an assumption, but the price to pay is that the KL divergence has erratic behaviours in many cases, starting with Dirac measures.
> Indeed, a KL divergence between two Dirac measures with disjoint supports is infinite.
> Thus, Lipschitzness helps to handle a broader range of prior/posterior pairs. However, we agree that the Lipschitzness requirement is a clear limitation (as it is to a wide variety of generalization bounds with similar assumptions) and we will make this more clear in the next version.
>
> **Discussion on Theorem 1**
>
> Previous PAC-Bayes bounds using supermartingales techniques implied that order 2 moments of the prior *and all considered posteriors* have to be uniformly bounded by a certain $C$. This suggests that those two distributions have to be robust enough to compensate for heavy-tailed data. For instance, if the data distribution of a point $z$ only admits a finite variance and that we consider the loss $\ell(h,z)= |h-z|$ where both $h,z$ lie in the real axis. Then, assuming $\mathbb{E}[z^2]\le V$, then by Cauchy Schwarz, we have  $\mathbb{E}[\ell(h,z)^2] \le \mathbb{E}[h^2] + 2V\mathbb{E}[|h|] +V^2 \le C$, the last inequality being our assumption. This invites us to consider posterior distributions with first and second moments satisfying the previous inequality, which is restrictive. On the contrary, our work assumes only such an assumption on the prior distribution: we are allowed to move freely on the space of distribution to find the best posterior. We thank you for this discussion which will enhance our manuscript. We will provide more detail on the high-level ideas of the proof in the next version to explain the use of the 'supermartingale toolbox'.
>
> **Role of $K$**
>
> We thought the role of $K$ with respect to the notion of generalisation, i.e., having multiple data-dependent priors may tighten the generalisation bound and thus, ensure good theoretical generalisation ability. In this work, we consider regularisation as directly incorporating the generalisation error into the algorithm. Understanding more deeply the role of $K$ into the optimisation procedure is a promising lead for future works.
>
>  **Online setting**
>
> The online counterpart of the classical bound Eq. (3) would involve a KL divergence at each time step while our approach Eq. (4) implies a Wasserstein distance. The important point is that we perpetrate the spirit of Haddouche and Guedj (2022), which showed that the natural online counterpart of the classical PAC-Bayes bound Eq. (3) comes with sound theoretical guarantees. Our approach is similar here: in Theorems 1 and 2, we show that our batch PAC-Bayesian algorithm is derived from sound theoretical bounds, and we show in Theorems 3 and 4 that it is the same for its online counterpart. Concerning the Dirac case, it is indeed a restriction for classical bounds Eq. (3). However, we focused on this important case as it cannot be handled properly by classical PAC-Bayes algorithms as the KL divergence of two Dirac with disjoint supports is infinite. In other words, our framework **does not** require Dirac priors (many other options can be used), but it can actually accommodate Dirac priors as opposed to KL-based bounds. An important message of our work is that the PAC-Bayes learning is rich enough to explain the generalisation ability of deterministic predictors (and not only probabilistic ones).
>
> **About the experiments**
>
> As suggested, we will include more discussion about the experiments in the next version. We are happy to provide supplementary insights into our experimental conclusions. We also provided new experiments with classical regularisers (see general answer).
>
> We thank you again for your review and are happy to include the above comment in the revised version of our work, and hopefully, the camera-ready.

---

> > ### Comment · Reviewer_gr5G · 2023-08-11
> >
> > Thank you for your thoughtful response to my comments.

---

> > > ### Author Response · Authors · 2023-08-17
> > >
> > > We would like to thank you again for replying quickly to our rebuttal and for your review.

---

### Author Rebuttal · Authors · 2023-08-09

We warmly thank all reviewers for their insightful reviews of our work. We are encouraged to see that our work generated enthusiasm and we answer thoroughly to all of the concerns raised by the reviewers. We address, in this general response, remarks raised by at least two reviewers.


**New experiments to study ERM with classical regularisation methods.**

Additional experiments have been asked by reviewers jPqk and tL5n to see the performance of classical regularisation methods such as the 'distance to initialisation' $||\mathbf{w}-\mathbf{w}\_0||$ and the weight decay $||\mathbf{w}||$. We point out that the first is a particular case of Algorithm 1 when $K=1$ (i.e., we treat the data as a single batch, and the prior is the data-free initialisation); the results are in Table 3 of Appendix C.4. More precisely, this table shows that, on most of the datasets, considering data-dependent priors leads to sharper results. This shows the efficiency of our method compared to the 'distance to initialisation' regularisation. Furthermore, we ran experiments with the weight decay regularisation which is gathered in the pdf document updated with the rebuttal.
This experiment demonstrates that on a few datasets (namely SENSORLESS and YEAST) when our predictors are neural nets, the weight decay regularisation fails to learn while ours succeeds as shown in Tables 1 (of our work) and 4 (of the new pdf).
More generally, the weight decay regularisation outperforms our method 3 times over 11 for batch learning with neural nets.

 **Clarity of Section 4.1**

Several comments have pointed out the lack of clarity in the first paragraphs of Section 4.1. We worked on the following points that will appear in the revised version of the paper:

1. We move the second paragraph of the 'Batch algorithm' part before Eq. (5). This makes the loss function we introduced appear more clearly, as suggested by Rev. tL5n.

2. Also, when our predictor space $\mathcal{H}$ is parametrised over $\mathbb{R}^D$ (for a certain $D>0$) equipped with the Euclidean distance $||.||$, we clearly define the distance $d$ between two predictors $h\_{\mathbf{w}},h\_{\mathbf{w}'}\in \mathcal{H}^2$ to be $d(h\_{\mathbf{w}},h\_{\mathbf{w}'})= ||\mathbf{w}-\mathbf{w}'||$. In Eq. (5), we also replace $||\mathbf{w}-\mathbf{w}\_i||$ by $d(h\_{\mathbf{w}},h\_{\mathbf{w}\_i})$ for the sake of consistency. This might clarify the concern of Rev. mMbV on how this Euclidean distance appeared in Eqs. (5),(6),(7) but raises the question of whether our losses are Lipschitz with regard to $\mathbf{w}$ and not $h_{\mathbf{w}}$. As Lemma 8 provided a positive answer for the linear model, we provided a supplementary lemma (with its proof) stating that our neural networks, as long as the weights are clipped (with an arbitrarily high constant), are Lipschitz with respect to their weight, explaining why our learning procedure is consistent with our theorems. As we considered a high value of the clipping, it did not interfere with our optimisation procedure. Those points will be clearly stated in the next version.

Again, we thank the reviewers for their work and hope this, as well as individual responses, clarifies all existing concerns.

---

### Decision · Program_Chairs · 2023-09-21

**Decision:**

Accept (poster)

**Comment:**

The paper introduces PAC-Bayesian learning with Wasserstein distance replacing the KL divergence in the metric space of hypotheses, presenting results for both batch and online learning scenarios. The generalization bounds are derived based on the Wasserstein distance between posterior and prior distributions over hypotheses, and the Lipschitz constant of the loss function. These bounds are applicable to heavy-tailed loss functions with bounded 2nd moments, and experiments with linear and neural network models demonstrate improved generalization compared to ERM and OGD on various datasets. Some weaknesses include variability in model performance based on batch size, and the need for further discussion on performance disparities between algorithms. The referees suggested reorganizing the paper.